# Cell-type-specific and disease-associated expression quantitative trait loci in the human lung

Heini M. Natri [1,12], Christina B. Del Azodi[2,3,12], Lance Peter[1], Chase J. Taylor [4], Sagrika Chugh[2,3,5], Robert Kendle[1], Mei-i Chung[1], David K. Flaherty[6], Brittany K. Matlock[6], Carla L. Calvi[4], Timothy S. Blackwell[4,7,8], Lorraine B. Ware [4,9], Matthew Bacchetta [10], Rajat Walia [11], Ciara M. Shaver[4], Jonathan A. Kropski [4,7,8,13], Davis J. McCarthy [2,3,5,13] & Nicholas E. Banovich [1,13] ✉

Common genetic variants confer substantial risk for chronic lung diseases, including pulmonary fibrosis. Defining the genetic control of gene expression in a cell-type-specific and context-dependent manner is critical for understanding the mechanisms through which genetic variation influences complex traits and disease pathobiology. To this end, we performed single-cell RNA sequencing of lung tissue from 66 individuals with pulmonary fibrosis and 48 unaffected donors. Using a pseudobulk approach, we mapped expression quantitative trait loci (eQTLs) across 38 cell types, observing both shared and cell-type-specific regulatory effects. Furthermore, we identified disease interaction eQTLs and demonstrated that this class of associations is more likely to be cell-type-specific and linked to cellular dysregulation in pulmonary fibrosis. Finally, we connected lung disease risk variants to their regulatory targets in disease-relevant cell types. These results indicate that cellular context determines the impact of genetic variation on gene expression and implicates context-specific eQTLs as key regulators of lung homeostasis and disease.

Genomic and functional studies have the potential to reveal the genetic, molecular and cellular drivers of clinical phenotypes, laying the groundwork for the development of targeted interventions. Many disease-associated variants identified in genome-wide association studies (GWAS) are located in the regulatory regions of the genome and contribute to disease risk and progression by effecting changes in gene expression[1]. Combining genotype information with transcriptional profiles allows for the identification of genetic regulators of gene expression (that is, expression quantitative trait loci (eQTLs)). This approach has been widely applied to bulk RNA sequencing of primary tissues, providing insights into the tissue specificity of regulatory effects and contributing to our understanding of the mechanisms underlying complex traits[2]. However, cell type and context (for example, disease status) and the specificity of trait-associated SNPs poses a challenge to understanding the regulatory mechanisms that modulate disease risk and progression.

Single-cell RNA sequencing (scRNA-seq) has emerged as a powerful tool for the transcriptional profiling of individual cells and cell types, mitigating many limitations of bulk RNA-seq. Capturing scRNA-seq profiles and genome-wide genotype information from a population of individuals allows for the unbiased, cell-type-specific interrogation of variant effects on gene expression. This approach can enable the discovery of eQTLs that are specific to rare or disease-relevant cell types and eQTLs that have opposing effects in different cell types, all of

which could go undetected in bulk RNA-seq of heterogeneous tissues. These context-specific eQTLs are more likely to escape the purifying selection that limits mutations impacting ubiquitous eQTLs and are thus more likely to have roles in disease[3,4].

Interstitial lung diseases (ILDs) are chronic, progressive respiratory disorders characterized by the scarring of lung tissue accompanied by epithelial remodeling, loss of functional lung alveoli and accumulation of extracellular matrix[5]. Pulmonary fibrosis is the end-stage clinical phenotype of ILD. Pulmonary fibrosis remains incurable; the most severe form of pulmonary fibrosis (idiopathic pulmonary fibrosis (IPF)) leads to death or lung transplant within 3–5 years of diagnosis[5,6]. The pathogenesis and progression of IPF involve a complex interplay of predisposing factors, cell types and regulatory pathways[7,8]. GWAS and meta-analyses have identified 20 IPF-associated variants, and polygenic analyses suggest that a large number of unreported variants contribute to IPF susceptibility[9]. Some of these variants are eQTLs in bulk lung tissue; however, their cell-type-specific regulatory consequences have not been explored.

To investigate the genetic control of disease-related gene expression in pulmonary fibrosis, we generated scRNA-seq data from the lung tissue samples of 114 individuals (66 individuals with ILD and 48 unaffected donors). Combining these data with genome-wide genotype data, we mapped shared, lineage-specific and cell-type-specific *cis*-eQTLs across 38 cell types (Fig. 1a). We analyzed these data in conjunction with IPF and other GWAS summary statistics to uncover the regulatory mechanisms underlying ILD risk and progression. Using interaction models, we reveal disease-specific regulatory effects that further elucidate the mechanisms underlying disease biology.

## Results

### scRNA-seq of 114 lung tissue samples

To enable cell-type-level eQTL mapping, we generated scRNA-seq and genome-wide genotype profiles for 114 individuals, including 66 (58%) with ILD and 48 (42%) unaffected donors (Fig. 1a and Supplementary Table 1). The ILD lungs included samples from 39 individuals with IPF and 27 with other forms of pulmonary fibrosis, including sarcoidosis (*n* = 4), connective tissue disease-associated ILD (*n* = 3), idiopathic nonspecific interstitial pneumonia (*n* = 3), coal worker's pneumoconiosis (*n* = 3), chronic hypersensitivity pneumonitis (*n* = 2), interstitial pneumonia with autoimmune features (*n* = 2) and unclassifiable ILD (*n* = 10). Most (67%) the lung samples were from individuals with self-reported ethnicity of European ancestry; 53 (46%) reported past or present tobacco use (Fig. 1b).

Single-cell suspensions were generated from fresh peripheral lung tissue samples and processed using the 10X Genomics Chromium platform. For the 55 ILD lung samples, two libraries were prepared from differentially affected (more or less fibrotic) areas of one lung to account for regional heterogeneity. Genotype data was obtained through low-pass whole-genome sequencing (WGS) followed by imputation (Methods). We performed data integration, dimensionality reduction and unsupervised clustering of the 475,047 cells passing quality control using the Seurat package[10] (Methods and Supplementary Figs. 1–3). Based on marker gene expression (Supplementary Table 2), we identified 43 cell types with a median of 5,811 cells (minimum = 253, maximum = 94,413, mean = 11,048 cells; Fig. 1c).

### Most eQTLs are shared between cell types

Out of 43 annotated cell types, we selected 38 that had 40 or more donors with five or more cells for that cell type to use for eQTL discovery (Fig. 1d). These inclusion criteria were selected to maximize our ability to map eQTLs with confidence across many cell types (Supplementary Note 1). Pseudobulk eQTL mapping was performed on each cell type using LIMIX according to the optimized approach described in ref. 11. To maximize precision and overcome varying statistical power across cell types, we used multivariate adaptive shrinkage, a statistical method

for analyzing measures of effect sizes across many conditions to identify patterns of sharing and specificity[12]. After applying multivariate adaptive shrinkage with mashr (Methods), eQTLs were considered significant if they had a local false sign rate (LFSR) of 0.05 or less in at least one cell type and 0.1 or less in any additional cell type. A gene was considered an eGene for a cell type if any eQTL for that gene was significant. Of the 6,995 genes tested for eQTL (Methods), 6,637 (95%) were eGenes in at least one cell type. The number of eGenes found per cell type was greater for more abundant cell types (Fig. 2a), with a positive correlation ($R = 0.66$, $P = 6.6 \times 10^{-6}$) between the number of eGenes and the number of individuals used for mapping (Fig. 2b). To evaluate the robustness of these results, we used a permutation scheme by shuffling genotypes and repeating the analysis for each cell type, and then comparing the permuted $P$ values to the observed $P$ values and to a theoretical null distribution (Supplementary Figs. 7 and 8). We observed no notable deviation between the empirical and theoretical null distributions, demonstrating that our approach was well-calibrated to avoid false positives.

To summarize the overall pattern of eQTL sharing between cell types and compare this pattern with the transcriptional similarity, we visualized the top two principal components of the median pseudobulked gene expression levels across all 38 cell types for the 6,995 genes included in the eQTL mapping (Fig. 2c) and of the mashr-estimated effect sizes of top eQTL across all 38 cell types (Fig. 2d). This analysis demonstrated that the relationships between the regulatory mechanisms across lung cell types largely reflected the differences in expression patterns across cell types. We identified a set of top eQTLs by selecting the eQTL with the lowest, significant LFSR for each gene in each cell type. Using these criteria, there were 50,389 top eQTLs, with a median of 7 top eQTLs per gene across cell types (minimum = 1, maximum = 33). Top eQTLs were considered shared between two cell types if they were significant in both cell types and their mashr-estimated effect size was within a factor of 0.5. Across all cell types, the median pairwise sharing of top eQTLs was 93.5% (minimum = 55%, maximum = 99.3%; Fig. 3). The epithelial and endothelial lineages had the highest levels of interlineage sharing (median = 97.9%) while sharing between cell types within the mesenchymal lineage (median = 96.9%) and the immune lineages (median = 95.4%) was slightly lower.

We further classified top eQTLs as global (*n* = 34,030), multi-cell type (*n* = 14,027) or unique to a specific cell type (*n* = 2,332) (Methods). Global top eQTLs tended to be found in genes with higher average expression and that were more widely expressed across cells (Supplementary Fig. 10). Top eQTLs unique to a single cell type tended to have higher absolute estimated effect sizes (Supplementary Fig. 10), probably due in part to limited statistical power to detect cell-type-specific effects in some cell types (Supplementary Fig. 10). Finally, these cell-type-specific top eQTLs also tended to be located further from the transcription start site (TSS) (Supplementary Fig. 10) of their target, which is consistent with the observation that cell-type-specific eQTLs typically impact enhancers, while widely shared eQTLs impact promoters[13,14]. We overlapped the top eQTLs with genic annotations from TxDb. Out of the 63% of sc-eQTL SNPs (eSNPs) that overlapped genic annotations, 7.9% were located on promoters and 30.3% were intergenic; the remaining overlapped at least one intron, exon or UTR. Out of the sc-eQTLs unique to a single cell type, shared between multiple cell types or globally across all cell types, 4.0%, 7.1% and 6.7% were located on promoters, and 14.2%, 26.0% and 22.9% were intergenic, with no statistically significant differences in annotations between eQTLs belonging to the different categories (Supplementary Fig. 13a). We further explored the overlap of the various classes of eQTL among all enhancers in the EnhancerAtlas 2.0 (ref. 15) lung tissue enhancers, and the human lung epithelial cell line (Calu-3) enhancers, as well as *cis*-regulatory elements in the Human Cell Atlas[16]. Testing for the equality of proportions overlapping enhancer annotations between eQTLs and the null set, we found that multistate sc-eQTLs were more likely to

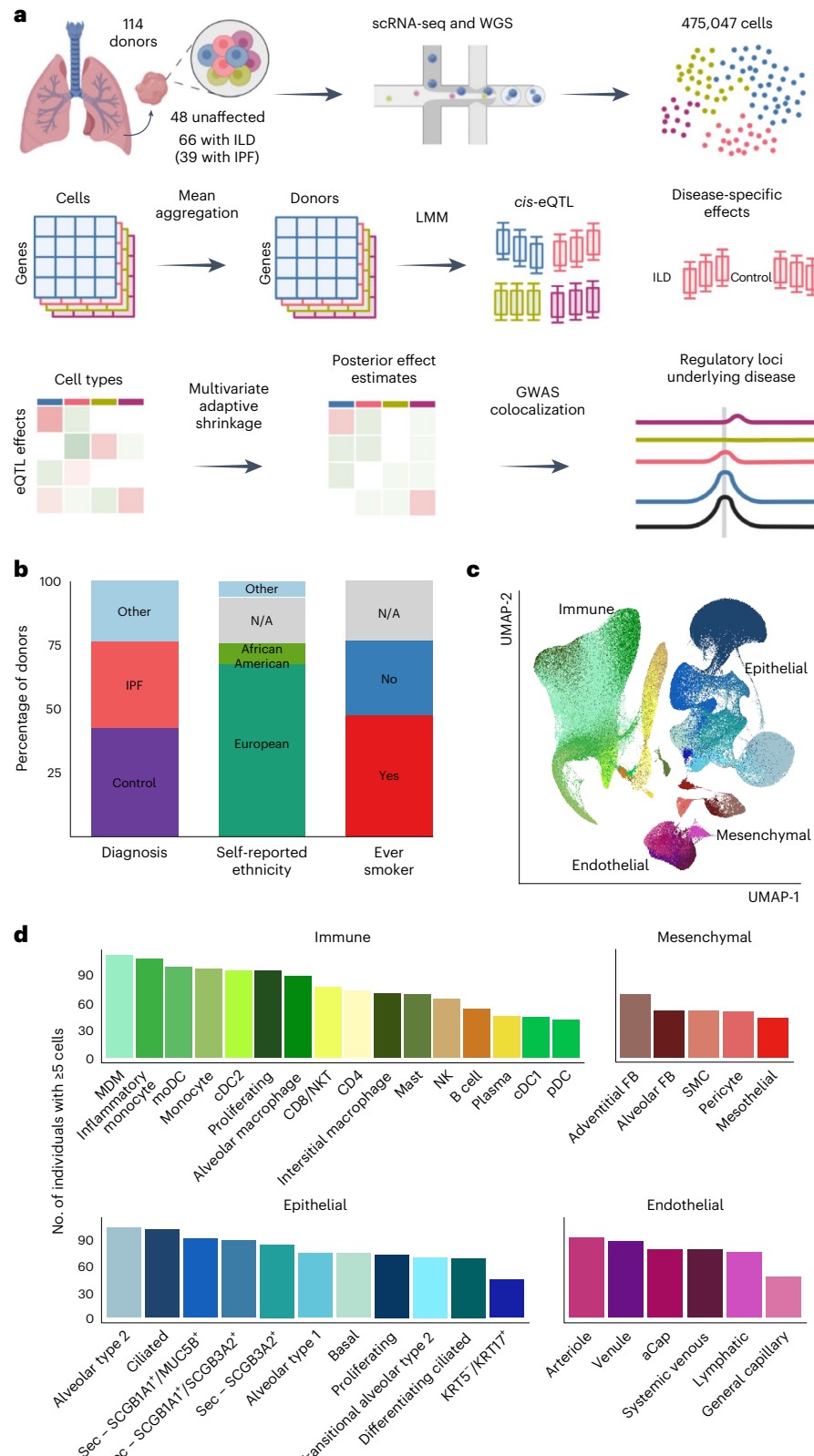

**Fig. 1 | Mapping eQTLs across cell types in the human lung. a**, Schematic illustration of the present study. **b**, Percentage proportions of donors according to diagnosis (42.1% unaffected controls, 34.2% IPF, 23.7% other ILD), self-reported ethnicity (66.7% European, 9.6% African American, 17.5% N/A, 6.1% other) and smoking history (46.5% ever smoker, 29.8% never smoker, 23.7% N/A). **c**, UMAP dimensionality reduction of 437,618 cells across the 38 cell types included in the eQTL analysis. Pseudocoloring indicates cell type; primary cell lineages are labeled. **d**, Numbers of donors with ≥5 cells for each cell type included in the analysis. LMM, linear mixed model; moDC, monocyte-derived dendritic cell; N/A, not applicable; NK, natural killer cell; NKT, natural killer T cell; pDC, plasmacytoid dendritic cell; SMC, smooth muscle cell. Panel **a** created with BioRender.com.

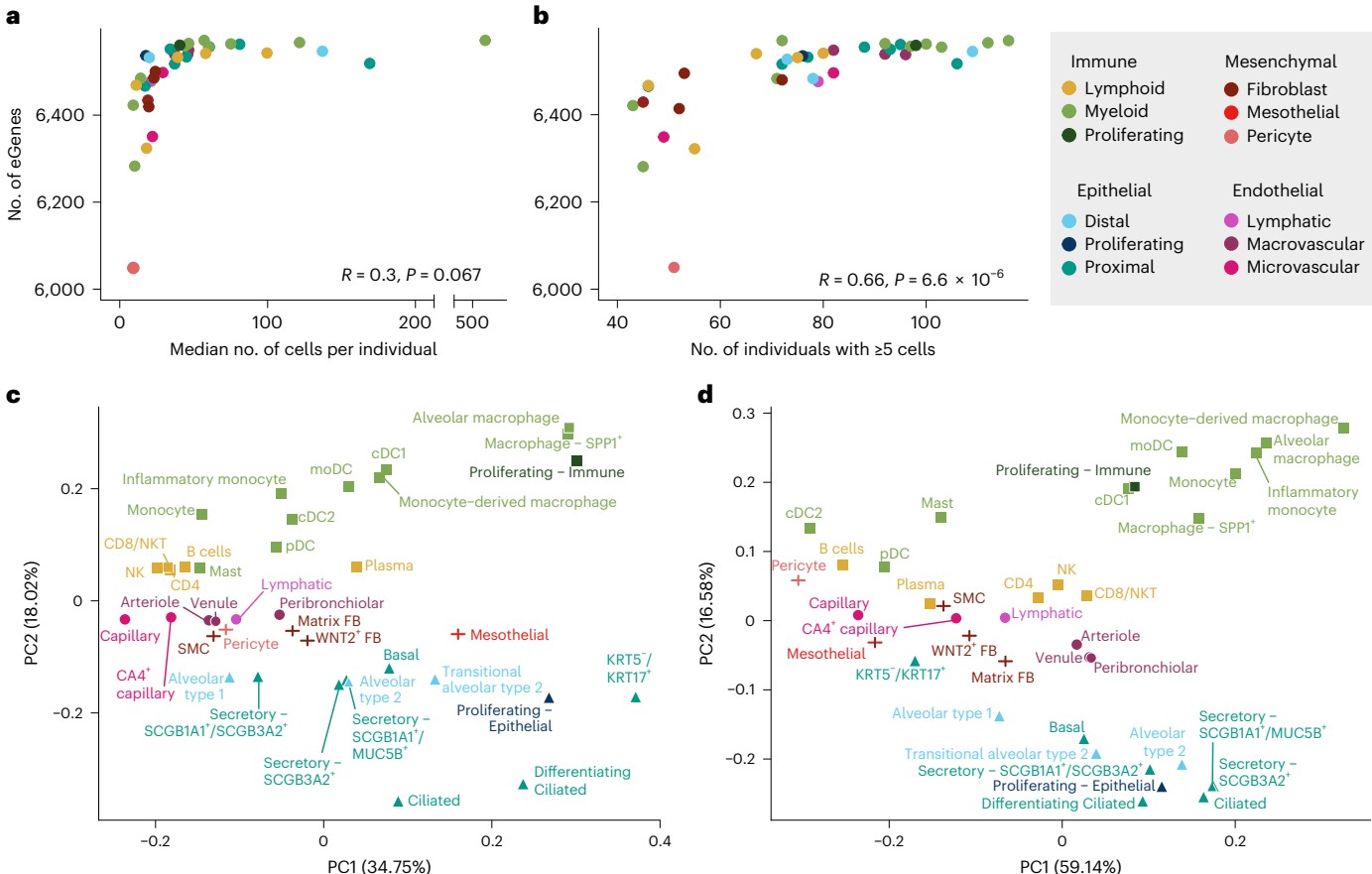

**Fig. 2 | sc-eQTL structure reflects lineage and cell type relationships.**
**a**, Comparison of the number of eGenes per cell type and the median number of cells per individual of that cell type (two-sided Pearson correlation). Cell types are colored according to sublineage. **b**, Comparison of the number of eGenes

per cell type and the number of individuals with at least five cells of that cell type (Pearson correlation). **c**, Principal component analysis (PCA) plot of pseudobulk expression across the 6,995 genes included in the eQTL mapping analysis. **d**, PCA plot of mashr-estimated effect sizes for the top eQTLs ($n = 50,389$).

be found overlapping the Human Cell Atlas *cis*-regulatory elements than the null set ($P = 3.502 \times 10^{-11}$; Supplementary Fig. 13b).

To explore the pattern of eQTL sharing across cell types more closely, we focused on multi-cell-type top eQTLs. We pruned these top eQTLs to get a representative sample for plotting ($n = 3,725$; Supplementary Table 5) and adjusted the sign of the effect sizes to where positive indicates the common effect direction and negative indicates an opposite effect direction; Methods). In an unsupervised clustering of the sign-adjusted effect sizes of these pruned eQTLs, we identified distinct classes of eQTLs (Fig. 4), including groups of eQTLs primarily active in epithelial or immune cell types, or exhibiting opposing effects between lineages. To connect these eQTLs to biological processes, we tested for the enrichment of their target eGenes among Gene Ontology (GO) terms against a set of 6,995 background genes (Fig. 4 and Methods). The eQTLs in cluster 3 were primarily active in the epithelial cell types and were enriched for genes involved in the regulation of JUN kinase, which has been implicated in lung fibrosis and is a potential target for interventions for ILD[17]. Epithelial eQTLs in cluster 5 were enriched for genes associated with metabolism and response to bacteria. The eQTLs in cluster 4 were primarily significant in the myeloid innate immune cell types and showed enrichment for genes involved in, for example, cholesterol metabolism. Furthermore, eQTLs in cluster 1 were mainly significant in the immune lineage and were enriched for genes contributing to cholesterol homeostasis, reflecting the central role of cholesterol metabolism in immune functions[18]. Cluster 7, also mainly active in the immune lineage, was enriched for genes involved with, for example, lipid transport. Lipid mediators have an important role in lung fibrosis[19]. The eQTLs in cluster 2 showed opposing effects

between the epithelial and immune lineages and were enriched for genes associated with highly lineage-specific functions, such as epithelial cell morphogenesis.

**Disease-specific eQTLs are highly cell-type specific**

To identify eQTLs specific to healthy or affected individuals or showing a different direction or degree of effect in the two groups, we performed disease-state interaction eQTL (int-eQTL) mapping (Methods). Testing across 33 cell types with five or more individuals with ILD and five or more unaffected donors and a minor allele frequency (MAF) ≥ 5% in each group, we detected 83,596 int-eQTLs. Applying this same analysis to our data after permuting the disease status resulted in 829 int-eQTLs, supporting a 1% false positive rate. Compared to the non-int-eQTLs, there was substantially less lineage and cell type sharing of int-eQTLs (Fig. 5a and Supplementary Fig. 12): for each gene, there was a median of 21 top int-eQTLs (minimum = 2, maximum = 28), resulting in a total of 75,482 top int-eQTLs. Compared to the top non-int-eQTLs, int-eQTLs were further from the TSS (mean absolute distance, sc-eQTL = 43.1 Mb, int-eQTL = 52.9 Mb, *t*-test $P = 2.22 \times 10^{-16}$) and had larger effect sizes (mean absolute mashr posterior beta, sc-eQTLs = 0.10, int-eQTLs = 0.66, *t*-test $P = 2.22 \times 10^{-16}$; Fig. 5b) and higher MAFs (mean MAF sc-eQTLs = 0.29, int-eQTLs = 0.37, $P = 2.22 \times 10^{-16}$). Some disease int-eQTLs were linked to overall expression differences between groups (Fig. 5c): 43% of int-eGenes were differentially expressed (adjusted $P < 0.1$) between ILD and unaffected samples in the particular cell type. Out of these genes, 50.8% were expressed at a higher level in ILD. However, 21% of int-eGenes were widely expressed (>30% of cells) in both groups in the particular cell

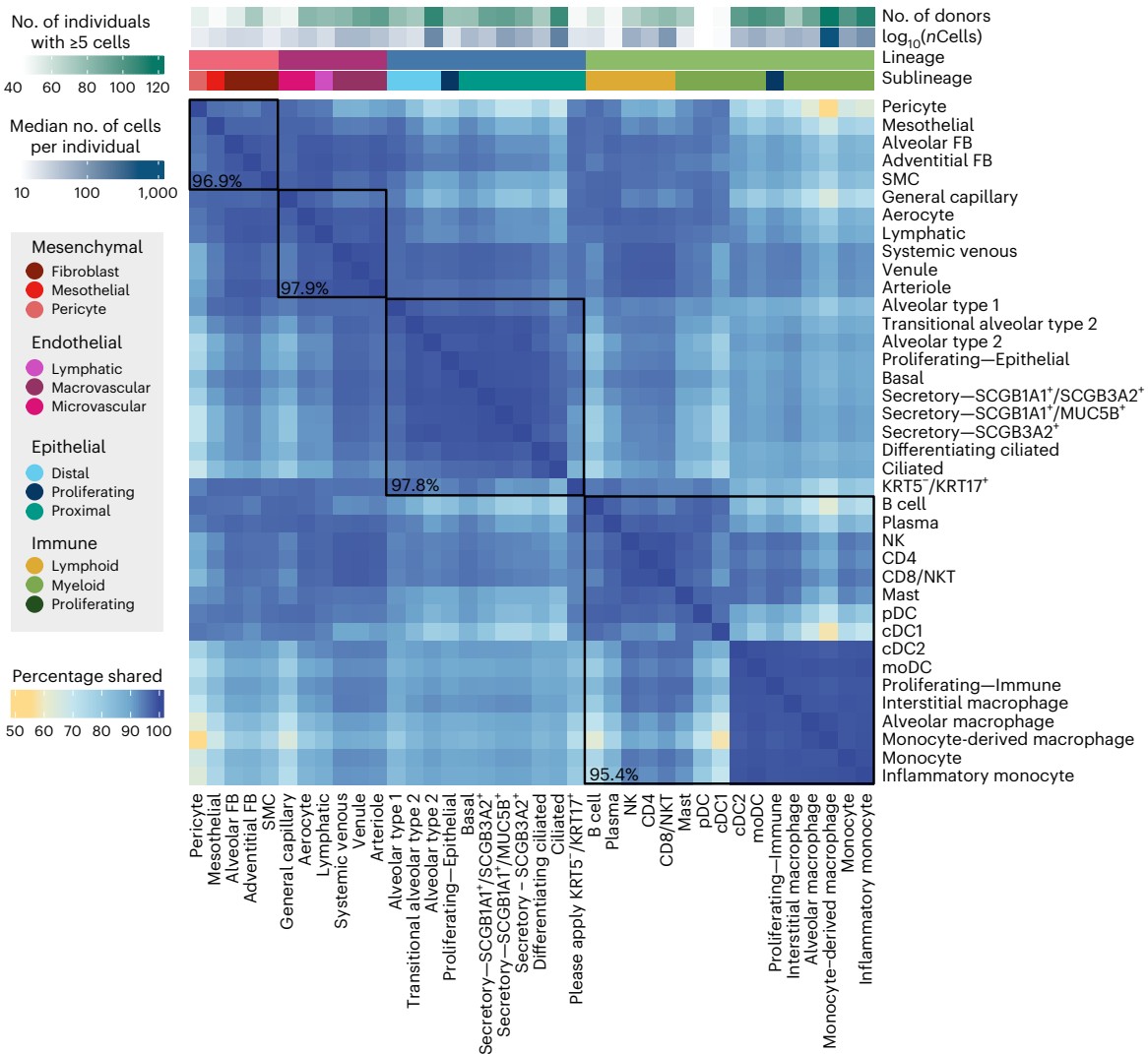

**Fig. 3 | eQTLs are largely shared between lung cell types.** Percentage of top eQTLs (*n* = 50,389) shared between two cell types. Top eQTLs are considered shared if they are significant in both cell types (LFSR ≤ 0.1) and the mashr-estimated effect size is within a factor of 0.5. Cell types are annotated above according to lineage, sublineage, the number of individuals with five or more cells and the median number of cells per individual for that cell type. Median pairwise percentage sharing per lineage is shown in black.

type and did not exhibit notable differences in expression levels (log fold change < 0.2), indicating that these eGenes were equally expressed but were differentially affected by *cis*-regulatory loci. These include *DSP* with three top int-eQTLs, including rs2003916, which was not significantly associated with IPF risk in the GWAS meta-analysis (*P* = 0.15) but showed differential effects between individuals with ILD and unaffected donors in four of the tested epithelial cell types (Fig. 5d).

To further interrogate the mechanisms underlying these int-eQTLs, we analyzed the int-eQTLs associated with eGenes expressed equally between individuals with ILD and unaffected donors for the enrichment of known transcription factor binding sites (TFBS) (Methods). We identified 42 significantly enriched transcription factor motifs (*q* < 0.05), including WT1, several SOX, HOX and PAX family members, ERG and NF1 (Fig. 5e and Supplementary Table 6). Several of these have known importance in lung fibrosis. WT1 functions as a positive regulator of fibroblast proliferation, myofibroblast transformation and extracellular matrix production[20]. A number of SOX transcription factors are upregulated in IPF and are associated with fibroblast activation[21,22]. Out of the 37 genes encoding transcription factors disrupted by int-eQTLs that were also tested for differential expression, 30 were differentially expressed between ILD and unaffected samples in at

least one cell type when using a significance threshold of adjusted *P* < 0.1. When examined across all cell types with significant differential expression, 43.0% of these genes were expressed at a higher level in the ILD samples. The seven that were equally expressed between cases and controls (adjusted *P* > 0.1), including WT1, SOX10, PAX7, HOXA11, HOXD12, NKX6-1 and SCRT1, could contribute to ILD pathogenesis through differences in protein levels or localization, differential binding to *cis*-regulatory elements or chromatin-level differences in addition to or instead of differential transcription factor abundance. We further examined the expression of these transcription factors by contrasting donors with 0/0 genotypes for rs2003916 (Fig. 5d) and those with at least one alternative allele or those with two alternative alleles. We found no differential expression of the significantly enriched transcription factors in any of the epithelial cell types included in the eQTL analysis, corroborating that the effect is not due to overall differences in transcription factor expression, but due to sequence-level or chromatin-level differences.

We assessed the level at which sc-eQTLs and int-eQTLs are replicated in bulk analyses by overlapping the eQTLs detected in this study with lung eQTLs from the Genotype-Tissue Expression (GTEx) project (Supplementary Note 2 and Supplementary Fig. 15)[2].

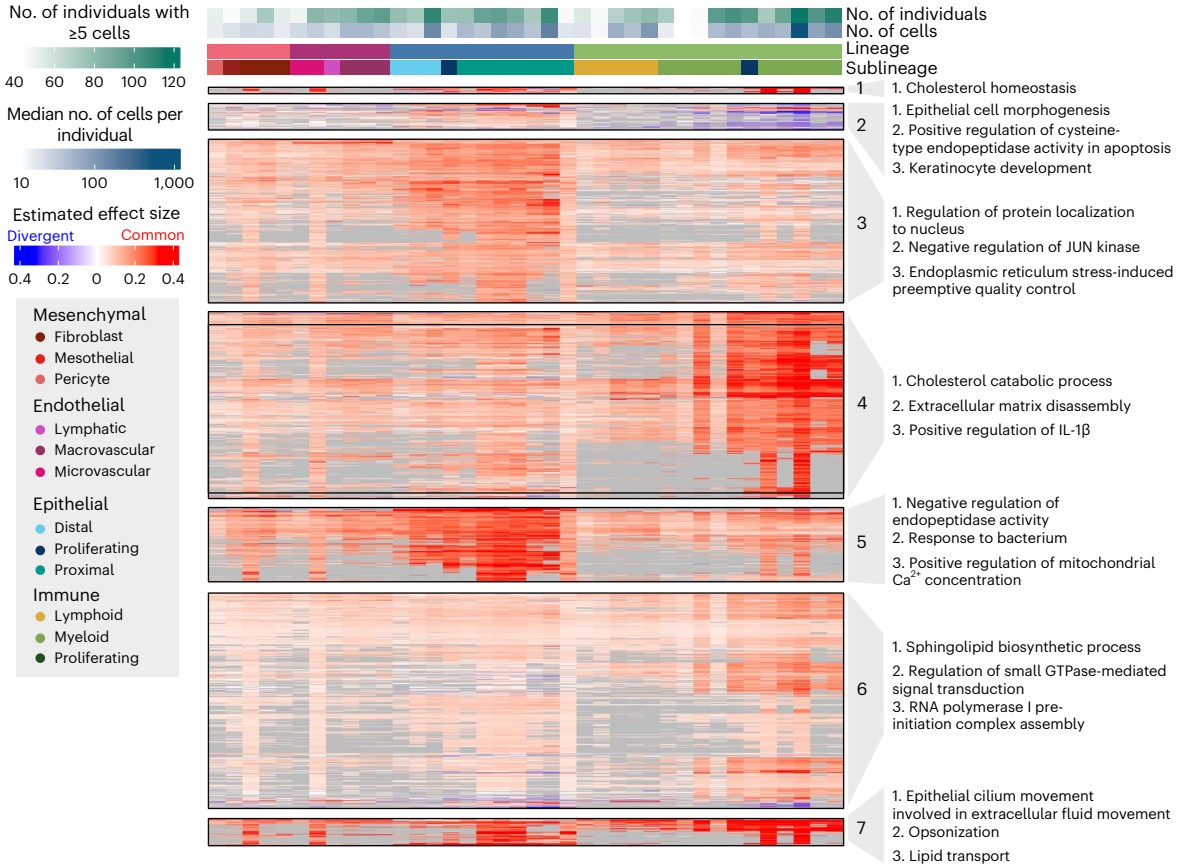

**Fig. 4 | Multi-cell-type eQTLs act in a highly lineage-specific manner.**
Visualization of a representative subset (Methods) of multi-cell-type top eQTLs and IPF-GWAS eQTLs ($n$ = 2,158). eQTLs are clustered according to their estimated effect sizes, with nonsignificant associations set to zero. eQTL effect sizes are not shown (gray) for genes expressed in less than 10% of cells of that cell type.

The most common effect direction for each eQTL is shown in red and cell types with opposite effect directions are shown in blue. The top three most significantly enriched GO terms for each cluster, excluding terms with support from less than two genes, are shown.

All classes of sc-eQTLs and int-eQTLs were enriched among GTEx lung eQTLs (Fisher's exact test, $P < 2.2 \times 10^{-16}$). Out of the globally shared and multi-cell-type top eQTLs, 19.1% and 21.9% were also eQTLs in the GTEx lung with a nominal $P < 1 \times 10^{-6}$ (Fig. 5f). However, only 11.7% of sc-eQTLs unique to a single cell type and 13.4% of int-eQTLs were GTEx-significant. This finding demonstrates the power of cell-type-specific and context-specific analyses in uncovering regulatory effects concealed by less granular approaches. We further compared the immune cell type eQTLs detected in this study to the ones reported in a previous study on peripheral blood mononuclear cells ($n$ = 982; Supplementary Note 2)[23]. Out of the 848 eQTLs for NK cells and 104 eQTLs for plasma cells detected by Yazar et al.[23] that were also tested for in our study, 31.0% and 19.2% were significant in our analysis of these cell types, respectively.

**Cell-type-specific patterns of colocalization at GWAS loci**

To connect the shared and cell-type-specific regulatory variants to IPF risk, we compared our results to a recent IPF GWAS meta-analysis[9]. All major classes of eQTLs were enriched among loci implicated (nominal $P < 1 \times 10^{-6}$, Supplementary Table 7) by the IPF GWAS meta-analysis (Fisher's exact test, globally shared $P < 5.09 \times 10^{-64}$, multi-cell-type $P < 1.83 \times 10^{-98}$, unique to a single cell type $P = 0.0525$), while a null set of nonsignificant eQTLs with a matched distribution of distances to the TSS was not ($P = 1$). GTEx bulk lung eQTLs were similarly highly enriched ($P = 2.22 \times 10^{-111}$) among the IPF GWAS loci. Surprisingly, disease interaction eQTLs were not more enriched among IPF GWAS loci than a null set of nonsignificant eQTLs.

In addition to the intersection analysis described above, we colocalized eQTL signals for 2,092 genes, including the target genes of the multistate eQTLs in Fig. 4 and 103 GWAS-implicated genes, with the IPF GWAS meta-analysis[9], the UK Biobank (UKBB) IPF GWAS[24] and an East Asian IPF GWAS[25] (Methods). We identified five loci with evidence of colocalization (posterior probability for a single shared causal variant greater than 0.6) between risk loci and eQTLs in at least one cell type. These patterns largely overlapped between the IPF GWAS meta-analysis and the UKBB (Fig. 6 and Supplementary Table 8). Three of these loci were eQTLs for genes previously implicated in a GWAS in the National Human Genome Research Institute (NHGRI)-EBI GWAS Catalog[26]: *MUC5B*, *DSP* and *KANSL1*. The locus associated with *KANSL1* in both the GWAS and eQTL analysis was also associated with the expression of *KANSL1-AS1* across several cell types in our dataset. Additionally, we found that an eQTL for the gene *JAML* was significantly colocalized with a locus from the GWAS analysis. This variant did not meet the criterion for genome-wide significance in the GWAS analysis but was an eQTL across a number of myeloid lineage cell types (Supplementary Fig. 19). *MUC5B* was robustly expressed and colocalized with the IPF GWAS meta-analysis and the UKBB IPF GWAS in SCGB1A1+/MUC5B+ and SCGB3A2+ secretory cells, implicating these as the most likely cell types in which the risk variant functions (Supplementary Figs. 17 and 18). In contrast to the mostly European IPF GWAS meta-analysis and UKBB, the *MUC5B* eQTL did not significantly colocalize with the East Asian IPF GWAS in any cell type, probably because of the low frequency of the risk allele in Asian populations[27]. The pattern of population sharing was different for the *DSP* eQTL, which was colocalized with the IPF GWAS

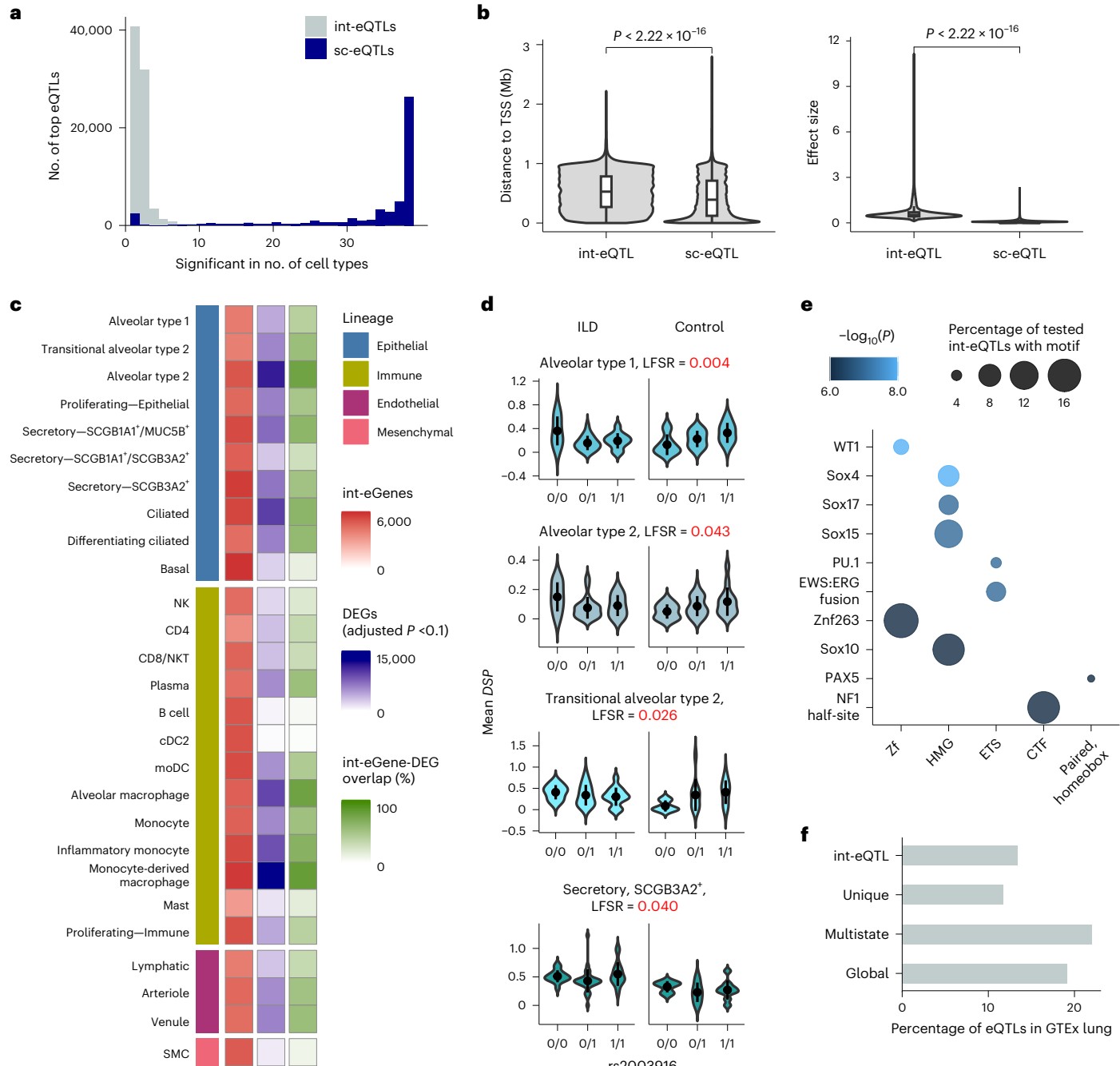

**Fig. 5 | Disease interaction eQTLs converge on pathways relevant to lung fibrosis. a**, Histogram of the cell type sharing of the top int-eQTLs and the top non-int-eQTLs. **b**, Comparison of absolute distances to the eGene TSS and absolute effect sizes of the top sc-eQTLs ($n = 50{,}506$) and int-eQTLs ($n = 83{,}596$). Two-sided $t$-test $P$ values are indicated. In the box plots, the lower and upper hinges correspond to the first and third quartiles. The upper whisker extends from the hinge to the largest value no further than 1.5 times the interquartile range (IQR) from the hinge; the lower whisker extends from the hinge to the smallest value at most 1.5 times the IQR of the hinge. **c**, Numbers of int-eGenes

and differentially expressed genes (DEGs) between fibrotic and unaffected samples, and proportion of their overlap for each cell type included in the int-eQTL analysis. **d**, Example of an int-eQTL for *DSP*. In the violin plots, the mean and two s.d. are indicated. **e**, Top transcription factor motifs enriched among int-eSNPs associated with eGenes that were equally expressed between individuals with ILD and unaffected donors but exhibited differences in eQTL effect sizes. Transcription factors are grouped according to family on the $x$ axis. **f**, Percentage of int-eQTLs, sc-eQTLs unique to a single cell type, multi-cell-type sc-eQTLs and globally shared sc-eQTLs that are also eQTLs in GTEx lung ($P < 1 \times 10^{-6}$).

meta-analysis in alveolar type 2, transitional alveolar type 2 and alveolar type 1 cells, and with the UKBB and the East Asian IPF GWAS in alveolar type 2 cells (Supplementary Fig. 21). The eQTL for *KANSL1* colocalized with the meta-analysis and UKBB in ciliated epithelial cells. Additionally, the eQTL for *KANSL1-AS1* antisense RNA was widely colocalized with the meta-analysis and UKBB across epithelial, immune and endothelial cell

types. However, the expression levels and eQTL effect sizes of *KANSL1* and *KANSL1-AS1* were highly correlated (Supplementary Figs. 22–24); both genes were ubiquitous but lowly expressed across cell types, impeding an exact evaluation of the cell type specificity of these effects.

When examining how these signals were colocalized in the bulk eQTL analyses, we found that the colocalization patterns of

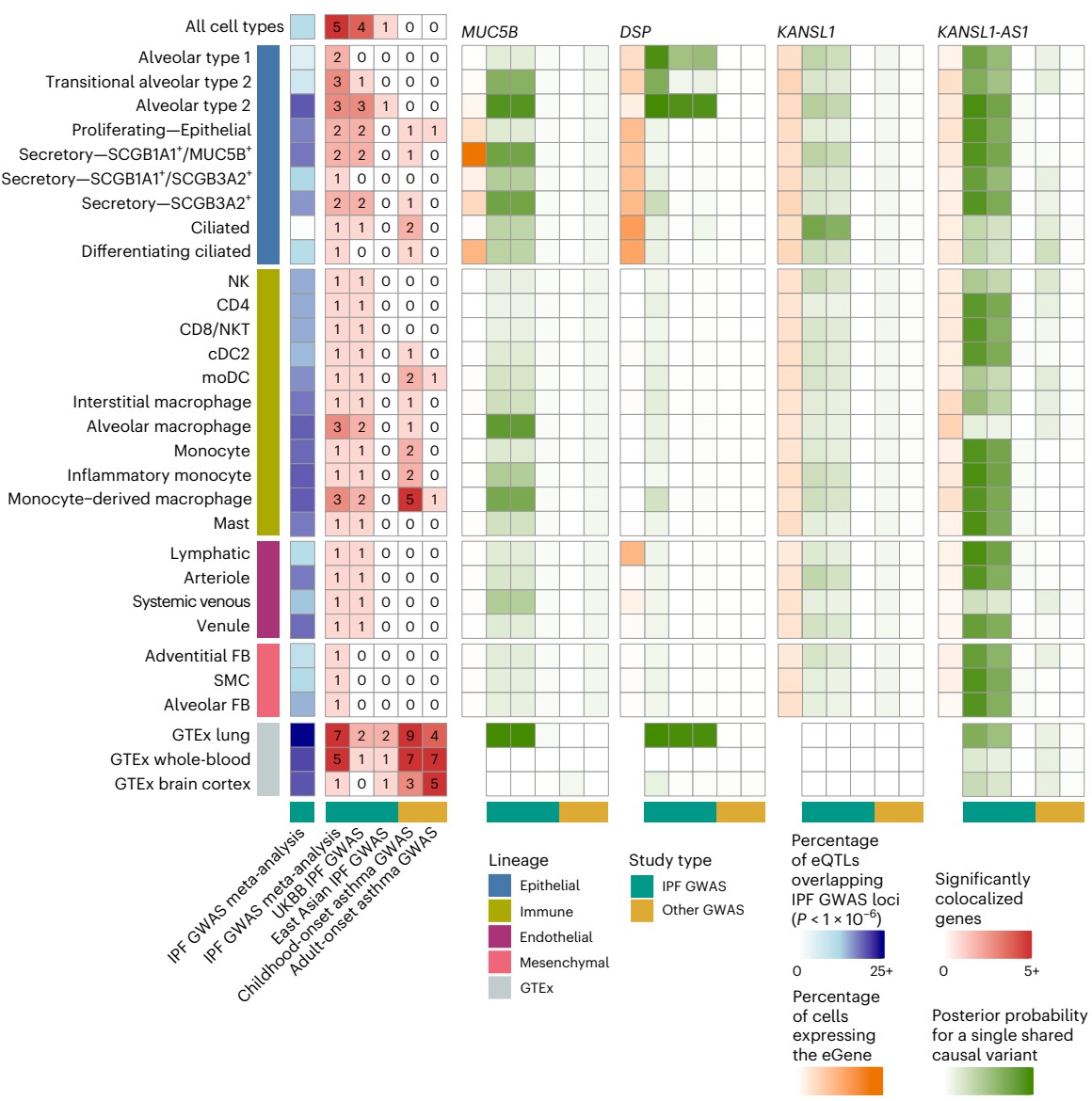

**Fig. 6 | Cell-type-specific eQTLs colocalize with the lung trait GWAS.** Numbers of SNPs that were nominally significant ($P < 1 \times 10^{-6}$) in the IPF GWAS meta-analysis and also eQTL (blue), the numbers of significant colocalizations between cell type and bulk eQTLs and three IPF GWAS, as well as childhood-onset and adult-onset asthma GWAS (red). Shown are the proportion of cells expressing the gene (orange) and the posterior probabilities for a single shared causal variant between the tested cell types and the GWAS for the selected top IPF-associated genes (*MUC5B, DSP, KANSL1, KANSL1-AS1*, shown in green) across 27 cell types with at least one colocalized gene.

*MUC5B* and *DSP* between GTEx lung and IPF GWAS reflected those of the cell-type-level analysis (Supplementary Fig. 15 and Supplementary Table 8). *MUC5B* was significantly colocalized with the IPF GWAS meta-analysis and UKBB, but not with the East Asian IPF GWAS. *DSP* was colocalized in all three IPF GWAS. *KANSL1*, however, did not colocalize between the GTEx lung and any IPF GWAS. To assess to what extent the genetic and cell-type-specific regulatory architecture of IPF risk may be shared with other lung diseases, we colocalized the cell-type eQTL signals with the childhood-onset and adult-onset asthma GWAS[28]. The childhood-onset asthma colocalization revealed a regulatory architecture distinct from IPF, with a lack of colocalization in epithelial cells and most of the significant colocalizations being specific to immune cells, particularly monocytes and monocyte-derived macrophages, which may shape some of the clinical and inflammatory features of asthma[29,30]. These results highlight the broader utility of this dataset in the investigation of other lung traits and diseases.

## Discussion

In this study, we present a characterization of regulatory genetic variants across major cell types in the human lung, using scRNA-seq to identify eQTLs at cell-type resolution. In total, we characterized eQTLs across 38 different cell types identifying *cis*-eQTLs in over 6,000 genes. Building on bulk eQTL studies, such as the GTEx project[2], which sought to characterize differences in gene regulatory architecture across tissues, we used a multivariate adaptive shrinkage approach to robustly identify shared and specific eQTLs across cell types[2]. In addition to the majority of eQTLs that were shared across cell types, we identified thousands of eQTLs that were limited to a subset or single cell type. These eQTL classes were enriched among chronic lung disease GWAS loci and DEGs in fibrotic lungs, suggesting that context-specific gene regulatory mechanisms are important but yet, to date, largely unrecognized contributors to the mechanisms underlying chronic lung diseases.

Highlighting the power of this approach, we demonstrate that many of the eQTLs identified in this study were not eQTLs in bulk

data from primary lung tissue (Fig. 5f). This was particularly true of eQTLs limited to a single cell type (11.7% significant in bulk) and disease interaction eQTLs, which were far less likely to be shared across cell types (13.4% significant in bulk). Both of these classes of eQTLs tended to be further away from the TSS than global and multistate eQTLs suggesting that these loci may be disrupting enhancers rather than promoters (Figs. 4a and 5b). This observation would be consistent with the cell type specificity of these eQTLs and would distinguish them from eQTLs identified in bulk studies, which are strongly enriched for disrupting promoter regions. Indeed, some work suggested that common eQTLs (enriched near promoters) are less likely to have functional relevance[1,31,32]. In addition to being more distal from the TSS, cell-type-specific eQTLs tended to have larger effect sizes (Figs. 4a and Fig. 5b). At present, it is uncertain whether the difference in effect size is due to statistical power to identify these associations or if cell-type-specific eQTLs inherently exhibit larger effect sizes. As this class of eQTL is the least likely to benefit from the mashr[12] approach, it seems plausible that we only have statistical power to identify those with large effects. If this is the case, future single-cell eQTL studies with increased sample numbers and cell type representation from rare cell populations are likely to identify a substantial number of additional cell-type-specific and context-specific eQTLs.

Over the past 10 years, there has been an increased appreciation for the degree to which eQTLs may be context-specific, starting first with tissue type, then to functional and environmental contexts, and finally to cell type[23,33–39]. The results of this study suggest that sc-eQTL studies have the power to elucidate this context specificity and that they will better recover eQTLs associated with disease states or environmental perturbations because these effects are less likely to be shared across cell types within a tissue.

In addition to a general characterization of eQTLs in the lung, this study is uniquely positioned to explore the interplay between genetic variation and the molecular underpinnings of chronic lung diseases including pulmonary fibrosis. Focusing first on the known risk loci identified in various GWAS studies, we found eQTLs to be enriched among GWAS risk loci regardless of class (Fig. 6). These enrichments were similar to those found in the bulk eQTL analysis from the human lung (Fig. 6); however, using cell-type-level associations, we were able to partition the function of these risk variants into discrete cell types. Indeed, we found that risk variants were most likely to be eQTLs in alveolar type 2 cells, followed by a number of cells from the myeloid lineage, including both resident and recruited macrophages (Fig. 6 and Supplementary Table 7). Using a more formal colocalization analysis, we found four GWAS loci with strong support for a shared causal variant with an eQTL (compared to seven colocalizations in the bulk eQTL data), for which we identified the likely cell type in which these risk variants are acting (Fig. 6). Our findings align with recent insights into the cellular and regulatory drivers of ILD. Epithelial cell types have a central role in driving alveolar remodeling in IPF[40]. Indeed, in a GWAS colocalization analysis, we found that the top IPF risk variants flanking *MUC5B* and *DSP* regulated the expression levels of their targets in specific epithelial cell types.

In addition to assessing the effect of known risk loci on gene expression traits, we also more directly examined how genetic variation may alter key regulatory processes involved in disease. Turning back to the disease interaction eQTL analysis, enabled by the collection of a cohort consisting of both affected and unaffected individuals, we assessed how these context-specific eQTLs may further drive disease processes. Roughly half of the interaction eQTLs were driven by differences in overall mean expression between the disease-affected and control samples. In the case of disease-emergent expression difference (expression increased in the disease-affected samples), loci that further upregulate gene expression may propagate additional molecular dysfunction. Focusing on the set of interaction eQTLs with similar mean expression across disease-affected and control samples, we found the loci to be enriched for TFBS associated with key biological processes related

to ILD. For example, we found enrichment for WT1 (ref. 20) and SOX family members[21,22], which previous experimental evidence connected to fibroblast activation and proliferation in the lung. The eQTLs that disrupt key binding sites probably further propagated the molecular dysregulation observed in ILD by modulating the binding efficiency of transcription factors and altering the expression of their direct and downstream target genes. Of note, int-eQTLs were not enriched for overlaps with risk variants, as anticipated based on the presumed requirement for disease-associated contextual cues for these variants to manifest their effects. We postulate that these context-specific eQTLs may have a role in disease progression rather than initiation. Again, these results highlight the importance of identifying context-specific eQTLs that are best captured using single-cell approaches.

Taken together, our study demonstrates the powerful application of single-cell genomics to study genetic regulation of gene expression in complex, solid, primary human tissues. Integrating scRNA-seq data from control and disease-affected lung samples with genetic data provides insights into the cell-type-specific function of risk variants for ILD and highlights int-eQTLs as a class of regulatory variants that contribute to disease pathobiology. Future work combining single-cell multiomic assays, healthy and disease-affected samples, and context-specific analysis methods, will be important to understand the interplay of dysfunctional genetic regulation and cellular contexts in complex human disease.

## Online content

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

[1]Translational Genomics Research Institute, Phoenix, AZ, USA. [2]St. Vincent's Institute of Medical Research, Melbourne, Victoria, Australia. [3]Melbourne Integrative Genomics, University of Melbourne, Melbourne, Victoria, Australia. [4]Division of Allergy, Pulmonary and Critical Care Medicine, Department of Medicine, Vanderbilt University Medical Center, Nashville, TN, USA. [5]School of Mathematics and Statistics, Faculty of Science, University of Melbourne, Melbourne, Victoria, Australia. [6]Flow Cytometry Shared Resource, Vanderbilt University Medical Center, Nashville, TN, USA. [7]Department of Cell and Developmental Biology, Vanderbilt University, Nashville, TN, USA. [8]Department of Veterans Affairs Medical Center, Nashville, TN, USA. [9]Department of Pathology, Microbiology and Immunology, Vanderbilt University Medical Center, Nashville, TN, USA. [10]Department of Cardiac Surgery, Vanderbilt University Medical Center, Nashville, TN, USA. [11]Department of Thoracic Disease and Transplantation, Norton Thoracic Institute, Phoenix, AZ, USA. [12]These authors contributed equally: Heini M. Natri, Christina B. Azodi. [13]These authors jointly supervised this work: Jonathan A. Kropski, Davis J. McCarthy, Nicholas E. Banovich. ✉e-mail: nbanovich@tgen.org

## Methods

### Compliance with ethical regulations
This study was approved by the local institutional review boards (IRBs) (Vanderbilt IRB nos. 060165 and 171657; Western IRB no. 20181836). Written informed consent was obtained from all participants.

### Participants, samples and tissue processing
The scRNA-seq data presented in this article include previously published[41] and unpublished samples (Supplementary Table 1). Lung tissue samples were processed as described previously by Habermann et al.[8]. Briefly, ILD tissue samples were obtained from lungs removed at the time of lung transplantation at either the Vanderbilt University Medical Center (VUMC) or the National Thoracic Institute. Control tissue samples were obtained from lungs declined for organ donation either at the Donor Network of Arizona or VUMC. Tissue sections were taken from multiple peripheral (within ~2 cm of the pleural surface) regions in each lung. For ILD-affected lungs, representatively diseased areas were selected on the basis of preoperative chest computed tomography, while for control lungs, the most normal-appearing region was identified by gross inspection and selected for biopsy. For ILD-affected lungs, diagnoses were determined according to the American Thoracic Society/European Respiratory Society consensus criteria[42]. No statistical methods were used to predetermine sample sizes but inclusion thresholds were determined to maximize the ability to map eQTLs with confidence across many cell types (Supplementary Note 2). Studies were approved by the local IRBs.

Tissue samples were digested in either collagenase I/dispase II (1 μg ml$^{-1}$) or Miltenyi Multi Tissue Dissociation Kit using a gentleMACS Octo Dissociator (Miltenyi Biotec). Tissue lysates were serially filtered through sterile gauze, 100-μm and 40-μm sterile filters (Fischer). The resulting suspensions then underwent cell sorting using serial columns (Miltenyi MicroBeads, CD235a and CD45) or fluorescence-activated cell sorting at VUMC or the Translational Genomics Research Institute (TGen). CD45$^-$ and C45$^+$ populations were mixed 2:1 in samples processed at VUMC and used to generate the scRNA-seq libraries. At TGen, calcein acetoxymethyl was used to stain live cells; 10,000–15,000 live cells were sorted directly into the 10X reaction buffer and transferred to the 10× 5′ chip A (10X Genomics).

### scRNA-seq library preparation and next-generation sequencing
scRNA-seq libraries were generated using the 10X Chromium platform 5′ library preparation kits (10X Genomics) according to the manufacturer's recommendations and targeting 5,000–10,000 cells per sample. From 12 donors, multiple tissue samples were processed and libraries were generated from separate biopsies taken from the same lung to account for regional heterogeneity (Supplementary Table 1). Next-generation sequencing was carried out on an Illumina NovaSeq 6000 or HiSeq 4000. The resulting sequenced data were filtered to retain reads with a read quality greater than 3; CellRanger Count v.3.0.2 (10X Genomics) was used to align reads onto the GRCh38 reference genome.

### Data integration, clustering, cell type annotation and differential expression
scRNA-seq data were processed and analyzed using Seurat v.4 (ref. 10). CellRanger Count outputs were imported to create a Seurat object for each sample. The sample-specific objects were merged and the proportions of reads arising from mitochondrial genes were calculated for each sample. The merged object was filtered to retain samples with more than 1,000 identified features or less than 25% of mitochondrial reads.

Samples sequenced across 24 batches were integrated using reciprocal PCA (rPCA) as follows: the merged object was split by flowcell and the count data in each batch-specific object was normalized; variable features were identified for each object and integration features across objects were selected with SelectIntegrationFeatures(); data in each batch-specific object was scaled and underwent PCA dimensionality reduction using 2,000 variable features. rPCA integration was carried out using 3,000 integration anchors and four reference batches (6, 12, 18, 24). PCA dimensionality reduction on the integrated data was performed using 3,000 variable features. To determine the optimal number of principal components to identify neighbors and to construct the uniform manifold approximation (UMAP), we determined the difference between the variation explained by each principal component and the subsequent principal component and identified the last point where the percentage change was more than 0.1%. A shared nearest neighbor graph was constructed with $k = 20$; clusters of cells were identified using the modularity optimization-based clustering algorithm[43] implemented in Seurat v.4.

The resulting clusters were divided into four major cell subgroups based on marker gene expression: PTPRC$^+$ for immune cells; EPCAM$^+$ for epithelial cells; PECAM1/$^+$PTPRC$^-$ for endothelial cells; and PTPRC$^-$/EPCAM$^-$/PECAM1$^-$ for mesenchymal cells. Each subgroup-specific object underwent the same dimensionality reduction and clustering approach as described above. We removed doubles using a manual approach, as described previously[8,41], by identifying clusters of cells that expressed markers from multiple lineages[8,41]. Our previous work found this method to be more conservative than automated approaches. Indeed, when applying DoubletFinder v.2.0 (ref. 44) to one lineage (epithelial cells), DoubletFinder recovered 8,230 doublets (3.7%), whereas the marker-based approach identified 18,588 doublets (8.5%). After manual doublet removal and reclustering, subgroup-specific objects were further annotated for specific cell types based on known marker genes (Supplementary Table 2).

For differential gene expression testing, we used the R/presto implementation of the Wilcoxon rank-sum test (wilcoxauc)[45].

### Low-pass WGS, genotyping and imputation
Flash-frozen tissue in DNA/RNA Shield was homogenized using a bullet blender. Genomic DNA was extracted using the Zymo Quick-DNA/RNA Microprep Plus Kit. Library preparation and low-pass WGS were carried out at TGen or by Gencove (Supplementary Table 9). At TGen, libraries were prepared using PCR-free Watchmaker Kits (Watchmaker Genomics) with a 200-ng input. Genomes were sequenced on a NovaSeq system at low coverage (typically 0.4–1×). The resulting sequenced data were processed and imputed using Gencove's imputation platform.

### Pseudobulk cell type eQTL mapping
For eQTL mapping, cells with more than 20% of reads mapping to the mitochondrial genes were removed (466,989 cells remained). Mapping was only performed on cell types with at least 40 donors with at least 5 cells of that cell type (38 cell types met these criteria). Mitochondrial genes, genes encoding ribosomal proteins (downloaded from https://www.genenames.org/cgi-bin/genegroup/download?id=1054&type=branch), genes expressed in less than 10% of cells in the study and genes with a mean count across all cells less than 0.1 were excluded, resulting in 6,995 genes for eQTL mapping.

Pseudobulk *cis*-eQTL mapping was performed according to the guidelines by Cuomo et al.[11]. For each cell type, raw counts were normalized and log$_2$-transformed using scran[46] and mean-aggregated to get a single value for each gene for each donor for each cell type. Donors with fewer than five cells for a cell type were excluded from eQTL mapping for that cell type; only cell types with at least 40 donors matching this criteria were included (maximum donors = 113). Biallelic, autosomal SNPs were filtered to include SNPs with an MAF greater than 5%, Hardy–Weinberg equilibrium $P > 1 \times 10^{-6}$, and further pruned to remove highly correlated SNPs (--indep-pairwise 250 50 0.9) using plink2 (ref. 47), resulting in ~1.9 million SNPs. We tested for associations for SNPs within 1 Gb upstream and downstream of the gene body.

Linear mixed models were used to map *cis*-eQTLs using the LIMIX_qtl framework (https://github.com/single-cell-genetics/LIMIX_qtl)[48]. Expression levels for each gene were quantile-normalized to fit a normal distribution (--gaussianize_method). To control for unwanted technical effects, the first 20 cell-type expression principal components were regressed out before model fitting (--regress_covariates). To account for variance due to population structure, we included the identity-by-descent relationship matrix generated by applying plink2 --make-rel on the filtered SNP data as a random effect. To account for differences in cell type abundance across donors, we included the number of cells aggregated (1/*n*Cells) as a second random effect, using the random effect weighting approach described by Cuomo et al.[11]. Random effects were marginalized from the model using the low-rank optimization method (--low_rank_random_effect) described by Cuomo et al.[49].

## Joint cell-type eQTL analysis

Joint analysis of the LIMIX estimated effect sizes and their corresponding standard errors across all 38 cell types was performed using multivariate adaptive shrinkage in R (mashr v.0.2 (ref. 12)) according to the approach outlined in the 'eQTL analysis outline' vignette from the authors (https://stephenslab.github.io/mashr/articles/eQTL_outline.html). In this approach, a weighted combination of learned and canonical covariance matrices that describe patterns of eQTL sparsity and sharing across cell types is used as a prior for generating adjusted summary statistics. The data-driven covariance matrices were estimated from a subset of strong associations with an LFSR lower than 0.1 in at least one cell type (*n* = 487), calculated using adaptive shrinkage in R (ashr v.2.2 (ref. 50)). Default canonical covariance matrices were used, representing equal effect sharing across cell types, the top five principal components from the strong associations and extreme deconvolution matrices obtained from those principal components. The model was fitted to a random subset of 10,000 SNP–gene associations and then applied to all associations tested.

## Assessing significance, sharing and eQTL classification

The LFSR calculated by mashr was used to assess significance. To further reduce the impact of differential power on assessing sharing of eQTLs across cell types, if an eQTL was significant in one cell type (LFSR ≤ 0.05), then it would be considered significant in other cell types at a less stringent threshold (LFSR ≤ 0.1). An eQTL was considered shared in a pairwise comparison between two cell types if the eQTL was significant in both cell types and the estimated effect size was within a factor of 0.5. An eQTL was classified as global if it was significant in at least 36 of the 38 cell types (31 of 33 cell types for int-eQTLs). This two-cell-type buffer was included to reduce the impact of low-powered cell types on our categorization. eQTLs that were significant in only one cell type were classified as unique and eQTLs significant in 2–36 cell types (2–31 for int-eQTLs) were considered multi-cell-type eQTLs.

To simplify plotting of the top eQTLs (Fig. 4), a pruning step was included, where for each gene, if there was a single top eQTL, that eQTL was retained. If there were two top eQTLs, the Euclidean distance between the centered absolute values of the estimated effect sizes across cell types for the two eQTLs were compared. If the distance was greater than the set threshold (distribution = 0.2), both were retained. If the distance was less than the threshold then the one that was significant in more cell types was retained. Finally, if there were more than three top eQTLs, the pairwise Euclidean distance between the centered absolute values of the estimated effect sizes for each pair of top eQTLs was calculated. If all pairwise distances were above the threshold, all were retained. Otherwise, hierarchical clustering was performed and the tree was cut using cutree at a *k* between 2 and 5, which maximized the silhouette width. For each cluster, the top eQTL that was significant in most cell types was retained.

## Disease interaction cell-type eQTL mapping

To test for disease interaction eQTL effects, cell types were required to have at least ten control and ten ILD donors with at least five cells of that cell type, resulting in KRT5⁻KRT17⁺, pDC, cDC1, alveolar fibroblast and mesothelial cell types being excluded from the interaction eQTL analysis. SNPs were further filtered to remove those with an MAF < 5% in either the control or ILD donor populations (1.77 million SNPs remained). Interaction effects were tested using the run_interaction_QTL_analysis from LIMIX_qtl. Random effects were handled as described above for the eQTL mapping analysis. In the interaction term with SNP effect, we included the binary disease status (ILD versus unaffected). Fixed effects (for 20 principal components) were included but not regressed out before modeling because disease status was strongly correlated with some principal components. The results from this analysis were processed using mashr, with significance calling, as described above, for the eQTL analysis. For each cell type, we further pruned int-eQTLs to retain associations where the observed eSNP MAF for individuals with ILD and unaffected donors for the given cell type was greater than 0.05.

## Colocalization with GWAS and GTEx

Colocalization analysis was carried out between the cell type eQTL, GTEx lung eQTL and three IPF GWAS. The UKBB[24] and East Asian[25] IPF GWAS summary statistics were downloaded from the GWAS Catalog[26]. The discovery samples of these studies consisted of 1,369 cases with IPF, 14,103 cases with chronic obstructive pulmonary disease and 435,866 controls, and 1,046 cases with East Asian ancestry and 176,974 controls, respectively. Summary statistics from an IPF GWAS meta-analysis[9] leveraging data from three studies[51–53] were downloaded after gaining access by submitting a request (https://github.com/genomicsITER/PFgenetics)[54]. The meta-analysis consisted of 2,668 cases with IPF with European ancestry and 8,591 controls.

Additionally, GWAS on adult-onset and childhood-onset asthma[28] (26,582 adult cases with European ancestry, 13,962 child cases and 300,671 controls) were downloaded from the GWAS Catalog and included for comparison. For comparative analyses with bulk eQTL, GTEx lung, whole-blood and brain cortex eQTLs, summary statistics were downloaded from the GTEx Google Cloud bucket (https://console.cloud.google.com/storage/browser/gtex-resources)[55].

Bayesian colocalization analysis was performed using R/coloc v.5 (ref. 56). For the pseudobulk cell-type eQTLs, mashr LFSR was used in place of the nominal eQTL *P* value. A total of 2,092 genes, including the multi-cell-type eQTLs presented in Fig. 4 and 103 IPF GWAS variant flanking genes, were selected for the colocalization analysis; for each gene, colocalization testing was carried out between datasets that shared 100 or more variable (MAF > 0, <1) SNPs. Significantly colocalized loci were selected based on the posterior probability for a single shared causal variant of 0.6 or greater.

## Enrichment testing

We tested for the enrichment of the clusters of eQTLs in Fig. 4 among GO terms using a Fisher's exact test as implemented in R/TopGO v.2.46.0 (ref. 57). All genes included in the eQTL analysis were used as a background set. A *P* value threshold of 0.01 was used to select significant terms.

We used a Fisher's exact test to test for the enrichment of the various classes of sc-eQTLs (all eQTLs, globally shared, multistate, unique to a single cell type, *k*1–*k*7 in Fig. 4) among IPF GWAS risk variants. From the 1,617,891 SNPs tested for in the eQTL analysis and included in the IPF GWAS meta-analysis, a set of 473 GWAS variants was selected with a relaxed genome-wide nominal *P* value threshold of $1 \times 10^{-6}$. A null distribution of nonsignificant eQTLs was generated using the default rejection method of R/nullranges[58] v.3.16 to match the observed distribution of absolute distances to the TSS among the significant eQTLs.

To test whether the various classes of regulatory variants detected in the sc-eQTL analyses disrupted the binding of known transcription

factors, we used HOMER[59] v.4.11 to analyze eQTL positions for the enrichment of transcription factor binding site motifs. findMotifsGenome.pl with a default region size of 200 bp was used to detect enriched motifs. In each analysis, a null set of nonsignificant eQTLs with a matched distribution of distances to the TSS was used as a background. In the TFBS enrichment analysis of the int-eQTLs, the non-int-eQTLs were used as a background set. A $q$-value threshold of 0.05 was used to select significant motifs.

## Statistics and reproducibility
The statistical analyses are detailed in the Methods and figure legends and were performed using R v.4.1.1 and v.4.3.0.

## Reporting summary
Further information on research design is available in the Nature Portfolio Reporting Summary linked to this article.

## Data availability
Raw and processed 10X Genomics data, Seurat objects, mean-aggregated expression matrices and genome-wide LIMIX and mashr eQTL statistics can be found on the Gene Expression Omnibus under accession no. GSE227136. Genotype data are available on the database of Genotypes and Phenotypes under accession no. phs003521.

## Code availability
The code to reproduce the results presented in this study is available via Zenodo at https://doi.org/10.5281/zenodo.10459632 (ref. 60).

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

## Acknowledgements
We thank the Tennessee Donor Services and the Donor Network of Arizona and the patients and families who donated tissue samples to make these studies possible. This study was supported by a National Heart, Lung, and Blood Institute grant no. R01HL145372 and a Department of Defense award no. W81XWH1910416 to N.E.B. and J.A.K.; grant no. P01HL092870 to T.S.B.; grant no. K08HL136888 to C.M.S.; NHGRI grant no. R01HG011886 to N.E.B., D.J.M. and J.A.K.; the Doris Duke Charitable Foundation to J.A.K. and N.E.B.; National Health and Medical Research Council grant nos. GNT1195595 and GNT1162829 to D.J.M.; and National Institutes of Health grant nos. R01HL158906 and R01HL126176 to L.B.W. The Vanderbilt Flow Cytometry Shared Resource is supported by the Vanderbilt Ingram Cancer Center (P30 CA068485) and the Vanderbilt Digestive Disease Research Center (DK058404). Additional support was provided by the Vanderbilt Institute of Clinical and Translational Research (UL1 TR002243).

## Author contributions
N.E.B., D.J.M., J.A.K., H.M.N. and C.B.D.A. conceptualized the study. C.B.D.A. devised the methodology. H.M.N. and C.B.D.A. carried out the formal analysis. H.M.N., C.B.D.A., M.C., L.P., C.J.T., S.C. and R.K. carried out the investigation. N.E.B., D.J.M., J.A.K., L.B.W., R.W., T.S.B., C.M.S., D.K.F., B.K.M., M.B. and C.L.C. managed the resources. H.M.N., C.B.D.A. and L.P. curated the data. H.M.N., C.B.D.A., N.E.B., D.J.M. and J.A.K. wrote the original manuscript draft. H.M.N., C.B.D.A., N.E.B., D.J.M. and J.A.K. reviewed and edited the manuscript. H.M.N. and C.B.D.A. visualized the data. N.E.B., D.J.M. and J.A.K. supervised the study. N.E.B., D.J.M., J.A.K., L.B.W., D.K.F. and B.K.M. acquired the funding.

## Competing interests
J.A.K. reports advisory board fees from Boehringer Ingelheim, nonfinancial study support from Genentech and grant funding from Boehringer Ingelheim. N.E.B. reports consulting fees from Deepcell. L.B.W. has received advisory board fees from CSL Behring, Quark, Bayer and Merck, and has research contracts with Genentech and CSL Behring. T.S.B. reports consulting fees from Orinove, GRI Bio, Morphic and Novelstar Pharmaceuticals, research grants and contracts from Boehringer Ingelheim and Celgene, and nonfinancial study support from Genentech. R.W. reports consultant fees from Genentech and Boehringer Ingelheim. The remaining authors declare no competing interests.

## Additional information

**Correspondence and requests for materials** should be addressed to Nicholas E. Banovich.

nbanovich@tgen.org

# Reporting Summary

## Statistics

For all statistical analyses, confirm that the following items are present in the figure legend, table legend, main text, or Methods section.

| n/a | Confirmed | |
|---|---|---|
| ☐ | ☒ | The exact sample size (*n*) for each experimental group/condition, given as a discrete number and unit of measurement |
| ☐ | ☒ | A statement on whether measurements were taken from distinct samples or whether the same sample was measured repeatedly |
| ☐ | ☒ | The statistical test(s) used AND whether they are one- or two-sided *Only common tests should be described solely by name; describe more complex techniques in the Methods section.* |
| ☐ | ☒ | A description of all covariates tested |
| ☐ | ☒ | A description of any assumptions or corrections, such as tests of normality and adjustment for multiple comparisons |
| ☐ | ☒ | A full description of the statistical parameters including central tendency (e.g. means) or other basic estimates (e.g. regression coefficient) AND variation (e.g. standard deviation) or associated estimates of uncertainty (e.g. confidence intervals) |
| ☐ | ☒ | For null hypothesis testing, the test statistic (e.g. *F*, *t*, *r*) with confidence intervals, effect sizes, degrees of freedom and *P* value noted *Give P values as exact values whenever suitable.* |
| ☐ | ☒ | For Bayesian analysis, information on the choice of priors and Markov chain Monte Carlo settings |
| ☐ | ☒ | For hierarchical and complex designs, identification of the appropriate level for tests and full reporting of outcomes |
| ☐ | ☒ | Estimates of effect sizes (e.g. Cohen's *d*, Pearson's *r*), indicating how they were calculated |

*Our web collection on statistics for biologists contains articles on many of the points above.*

## Software and code

Policy information about availability of computer code

| Data collection | *Provide a description of all commercial, open source and custom code used to collect the data in this study, specifying the version used OR state that no software was used.* |
|---|---|
| Data analysis | Publicly available software used in data analysis:<br>CellRanger Count v3.0.2<br>R/Seurat v4<br>plink2 v1.07<br>LIMIX (https://github.com/single-cell-genetics/LIMIX_qtl)<br>R/mashr v0.2<br>R/ashr v2.2<br>R/coloc v5<br>R/TopGO v2.46.0<br>R/nullranges v3.16<br>HOMER v4.11<br><br>Custom scripts to reproduce the result presented here are available on GitHub at https://github.com/tgen/banovichlab/tree/master/ILD_eQTL. |

For manuscripts utilizing custom algorithms or software that are central to the research but not yet described in published literature, software must be made available to editors and reviewers. We strongly encourage code deposition in a community repository (e.g. GitHub). See the Nature Portfolio guidelines for submitting code & software for further information.

## Data

Policy information about availability of data

All manuscripts must include a data availability statement. This statement should provide the following information, where applicable:

- Accession codes, unique identifiers, or web links for publicly available datasets
- A description of any restrictions on data availability
- For clinical datasets or third party data, please ensure that the statement adheres to our policy

Raw and processed 10x Genomics data, Seurat objects, mean-aggregated expression matrices, and genome-wide LIMIX and mashr eQTL statistics can be found on GEO with the accession number GSE227136. Genotype data are available on dbGaP with the accession number phs003521.

## Human research participants

Policy information about studies involving human research participants and Sex and Gender in Research.

| | |
|---|---|
| Reporting on sex and gender | Self-reported gender information was available for 107 out of the 116 donors. Out of these, 29 reported female and 78 reported male. No sex-specific analyses were performed in this study due to the low number of female samples. |
| Population characteristics | Data were collected from 114 individuals, including 66 (58%) with ILD and 48 (42%) unaffected donors. The ILD lungs included samples from 39 individuals with IPF and 27 with other forms of PF, including sarcoidosis (n=4), connective tissue disease-associated interstitial lung disease (CTD-ILD, n=3), idiopathic nonspecific interstitial pneumonia (NSIP, n=3), coal worker's pneumoconiosis (CWP, n=3), chronic hypersensitivity pneumonitis (cHP, n=2), interstitial pneumonia with autoimmune features (IPAF, n=2), and unclassifiable ILD (n=10). The majority (67%) of the lung samples were from individuals with self-reported ethnicity information of European ancestry, and 53 (46%) reported past or present tobacco use. |
| Recruitment | Participants were be patients of the Clinical Investigator that are scheduled for a lung transplant surgery. The Clinical Investigator or a member of his research staff approached individuals to discuss the study and invite them to participate. |
| Ethics oversight | Studies were approved by the local Institutional Review Boards (Vanderbilt IRB nos. 060165 and 171657 and Western IRB no. 20181836). |

Note that full information on the approval of the study protocol must also be provided in the manuscript.

# Field-specific reporting

Please select the one below that is the best fit for your research. If you are not sure, read the appropriate sections before making your selection.

☒ Life sciences          ☐ Behavioural & social sciences          ☐ Ecological, evolutionary & environmental sciences

For a reference copy of the document with all sections, see nature.com/documents/nr-reporting-summary-flat.pdf

# Life sciences study design

All studies must disclose on these points even when the disclosure is negative.

| | |
|---|---|
| Sample size | The final dataset consisted of data collected from 114 donors, including 66 ILD and 48 unaffected donors. |
| Data exclusions | Donor VUILD65 was removed due to inconsistencies in metadata suggesting mislabeling. |
| Replication | sc-eQTL were compared with previously published datasets, i.e., GTEx. |
| Randomization | This is not relevant to the present study, as samples were not allocated to groups but comparisons were conducted between cases and unaffected donors. |
| Blinding | This is not relevant to the present study as samples were not allocated to groups. |

# Reporting for specific materials, systems and methods

We require information from authors about some types of materials, experimental systems and methods used in many studies. Here, indicate whether each material, system or method listed is relevant to your study. If you are not sure if a list item applies to your research, read the appropriate section before selecting a response.

## Materials & experimental systems

| n/a | Involved in the study |
|-----|----------------------|
| ☒ ☐ | Antibodies |
| ☒ ☐ | Eukaryotic cell lines |
| ☒ ☐ | Palaeontology and archaeology |
| ☒ ☐ | Animals and other organisms |
| ☒ ☐ | Clinical data |
| ☒ ☐ | Dual use research of concern |

## Methods

| n/a | Involved in the study |
|-----|----------------------|
| ☒ ☐ | ChIP-seq |
| ☒ ☐ | Flow cytometry |
| ☒ ☐ | MRI-based neuroimaging |

