## [Peer Review File · Nature Genetics]

Peer Review Information

Manuscript Title: Cell type-specific and disease-associated eQTL in the human lung

Corresponding author name(s): Professor Nicholas (E) Banovich

Editorial Notes:

Transferred manuscripts This document only contains reviewer comments, rebuttal and decision letters for versions considered at Nature Genetics.

Reviewer Comments & Decisions:

Decision Letter, initial version:

8th May 2023

Dear Professor Banovich,

hope this email finds you well.

I'm writing to let you know that your Article, "Cell type-specific and disease-associated eQTL in the human lung" has now been seen by 3 referees. You will see from their comments copied below that while they find your work of considerable potential interest, they have raised quite substantial technical concerns that must be addressed. In light of these comments, we cannot accept the manuscript for publication, but would be very interested in considering a revised version that addresses these serious concerns.

We hope you will find the referees' comments useful as you decide how to proceed. If you wish to submit a substantially revised manuscript, please bear in mind that we will be reluctant to approach the referees again in the absence of major revisions.

If you choose to revise your manuscript taking into account all reviewer and editor comments, please highlight all changes in the manuscript text file. At this stage we will need you to upload a copy of the manuscript in MS Word .docx or similar editable format.

*2) If you have not done so already please begin to revise your manuscript so that it conforms to our Article format instructions, available [here](http://www.nature.com/ng/authors/article_types/index.html). Refer also to any guidelines provided in this letter.

[redacted]

If you wish to submit a suitably revised manuscript we would hope to receive it within 6 months. If you cannot send it within this time, please let us know. We will be happy to consider your revision so long as nothing similar has been accepted for publication at Nature Genetics or published elsewhere. Should your manuscript be substantially delayed without notifying us in advance and your article is eventually published, the received date would be that of the revised, not the original, version.

Thank you for the opportunity to review your work.

Best wishes,
Chiara

Chiara Anania, PhD
Associate Editor
Nature Genetics
<https://orcid.org/0000-0003-1549-4157>

Referee expertise:

Referee #1: genetic epidemiology

Referee #2: lung disease - GWAS studies

Referee #3: human genetics - bioinformatics - systems biology

Reviewers' Comments:

Reviewer #1:

Remarks to the Author:

The authors present the results of single-cell RNA-seq data from lung tissue samples of 116 individuals (67 ILD and 49 unaffected donors) and show that cell specific expression data gives new insights into disease mechanism for ILD than from bulk eQTL data which is an important and novel result.

My main criticism of the paper is that many multi-dimensional patterns in the data are presented in very large and complicated heatmaps with extra information around the peripheries. I don't think these attempts to present "everything" in one plot help the reader get a clear impression of what the data is showing, and personally didn't tell me anything extra to what was said in the text, for example there is just way too much going on in Figure 4b. Could some of the heatmaps be replaced by scatter plots to show trends instead of having to try to compare shades of multiple colours across columns e.g. Figure 4c? Are the numbers feeding into the heatmaps provided in supplementary tables so that if one does happen to spot an interesting difference in shade the numerical values can be obtained?

In general there are too many figures (i.e. the multiple panels in each figure) not all of which are helpful.

Figure 1a and 1c should be separate Figures. 1b & 1d could be tables (odd that there are no tables in the manuscript at all).

Another concern is the conclusions around the different effect sizes and different distances to TSS sites between sc-eQTL and int-eQTL. The authors mention power effects but I am also wondering is how does the allele frequency spectrum compare between these 2 types of eQTL?

In Figure 4b how exactly were the top-eQTL "pruned" to give a representative sample?

I note some marginally significant P-values being presented as supporting conclusions, particularly in

Figure 4d (0.043, 0.04 - should be presented as same number of significant figures), is there really an important effect here? Also in Discussion "43% of int-eGenes were differentially expressed (adj. $p < 0.1$)", why such a lenient threshold here, what is the proportion using $p < 0.05$?

Figure 5b - is a t-test appropriate given the very different shapes of the distributions?

Difficult to keep track of what is meant by "top-eQTL". Top amongst what? Top for the cell type and gene? Can you be explicit - e.g. most significantly associated eQTL for the tissue/gene?

Supplementary table 1 only has 88 rows and yet it is referred to in a sentence mentioning 116 samples.

2nd paragraph page 5: "This demonstrated that the relationships between the regulatory mechanisms across lung cell types largely reflected the differences in expression patterns across cell types". Is this as expected? How do you explain the ones that have swapped position such as Mesothelial and CD8/NKT?

Also "Top eQTL are considered shared between two cell types if they are significant in both cell types and their mashr estimated effect size is within a factor of 0.5.", what is the rationale for this threshold? How sensitive are results to selection of this threshold?

2nd paragraph page 8: "Top eQTL were considered to be associated with IPF if the identified eGene was previously reported ($p < 1 \times 10^{-12}$ in an IPF GWAS meta-analysis⁹". Why this threshold?

Figure 5: DEG should be defined - presumably differentially expression gene?

Reviewer #2:

Remarks to the Author:

Natri and B Del Azodi et al. analyzed lung tissue single cell RNA sequencing data from IPF and controls and performed eQTL analysis using a pseudo-bulk approach to identify shared and cell type specific eQTLs. Authors describe the patterns of cell type specific eQTLs and identified disease-state interaction eQTLs that show differential regulation pattern by disease status. This study is a valuable resource for lung biology community for providing cell-type specific eQTL data. Pseudo-bulk approach is reasonable, but the study would be even more valuable if true single cell level heterogeneity could be incorporated in the analysis - such as eQTLs of cellular trajectories associated with alveolar epithelial regeneration and repair in this dataset with pulmonary fibrosis.

Major comments

1. Fig 4b: I believe the purpose of the figure is to demonstrate multi-cell type eQTLs tend to be lineage specific with similar effect sizes by lineage, but too much information is displayed which makes it less intuitive. Consider adding labels to the cluster figure as the text describes clusters by number, and focusing on a few representative, disease relevant eQTLs (that show concordant and discordant effects by cell types for example) in the main figure may be more helpful.

2. Disease-specific eQTL, such as DSP is very interesting that same locus has differential regulatory effects. The authors examined TF motifs and differential expression of TFs but it is not directly linked back to DSP. Was TFs differentially expressed among the different genotype groups by disease status? Was this variant associated with DSP or other IPF risk variants?

3. MUC5B GWAS variant colocalization is significant in AT2, transitional AT2, secretory cells and

macrophages while MUC5B is predominantly expressed in one secretory cell type (SCGB1A1+MUC5B+). As SCGB1A1+ secretory cells can transdifferentiate into other epithelial cell types in IPF, incorporating trajectory analysis and examining the dynamic eQTL pattern would further clarify the causal cell type.

4. For single-cell type eQTLs, how many of them were explained by cell-type specific expression vs. more globally expressed but showing specific regulatory effect on a single cell type?

5. KRT5-/KRT17+ cells are excluded from disease interaction analysis as it probably is not expressed enough in the non-diseased sample, however as it is a unique pathologic cell type in fibrosis, would be interesting to show the eQTL result/discussion on its findings. Similar to comment on 3 –trajectory analysis with AT2 cells and KRT5-/KRT17 will highlight the utility of single cell RNAseq.

Minor comments:

1. Providing the number of cells and relative abundance of cell types per subject and median number of genes in table S1 would be helpful.

2. Was regional heterogeneity incorporated in the analysis?

3. What covariates were used for the LIMIX? Smoking status will confound SNP-gene expression relationship, was it adjusted?

Reviewer #3:

Remarks to the Author:

Natri et al presents a large data set of lung single cell data, and map eQTLs in this single cell cohort. The work is timely and to my knowledge reflects the first single cell eQTL study in lung. However, I think there are a lot of technical concerns that I have. Single cell data can be sparse, and pseudobulk profiles may have variable sparsity if different numbers of cells are used to create profiles. I am worried that they may have a large number of false positives and may have inflated statistics. My major comments are below:

1. I think it would be useful to assess how well their results co-localize with GTEX lung. I think this would be a useful sanity check for their results, and would give a sense as to what is being missed in bulk analyses. Currently there is some mention of overlap in Figure 5f, but this may be confounded by thresholding. Specifically, I would like to know, for lead SNP-gene pairs out of their data, how well do effect sizes (betas) correspond to GTEX betas for the same SNP-gene pair. Also – for a given tissue, what percentage of discovered eGenes colocalize with lung GTEX results. Is GTEX capturing some cell-types better than others?

2. I had a few questions about the single cell analysis. The authors used shared—nearest neighbor strategy to perform batch correction. Given the tissue derived nature of their data set I would worry about batch effects. I could not find any discussion around the potential for batch effects in their data. Can the authors provide some metrics and reassurance that batch has been adequately addressed? Given the relatively small number of cells per cell-type per individual, batch effects may be a particular issue.

3. I noticed that the authors did build pseudobulk profiles from cell-types in individuals with as few as 5 cells. That strikes me as somewhat dangerous, given how sparse single cell data can be (as few as 1000 non-zero genes per their QC). I would expect that in these pseudobulk profiles that many zero count genes are present. I think the authors probably need to apply more stringent QC to their pseudobulk profiles. Given the number of cell types examined (43), and the number of individuals

(100) assayed, and the cells generated (500K), it means that on average pseudobulk profile is only constituted from ~100 cells. They should assess the number of zero count values for genes in their analysis. If a profile has too many zero count values, probably it should be removed. If there is too much sparsity in their data, authors may want to consider using coarser cell-type definitions.

4. Author's should use a computational strategy like scrublet to remove doublets. Currently it appears this is done qualitatively.

5. It isn't clear to me what type of SNP QC was done by the authors. Typically low quality imputed SNPs are removed.

6. Single cell eQTL maps are susceptible to statistical inflation. Given the relatively small number of individuals in their study (~100, with individual cell types having as few as ~50), and the potential for sparsity or outlier expression values to not work well with linear models, there is the potential for inflated p-values. To assess the possibility, I think it is important for authors to do permutations. The appropriate permutations here would be to reassign whole-genome genotype data to single cell data. E.g. each individual gets someone else's genome-wide genotype data. They should then carry through their statistical procedure. If robust, they should find very few if any eQTLs. While cumbersome – it is the only way to understand if they are producing false positives in their analysis.

7. To identify eQTL sharing, the authors identify the top SNP-gene pair for each cell-type and then assess consistently across cell types. This approach might suffer from thresholding of effects. One alternative strategy that might be more accurate might be to map eQTLs across all celltypes (e.g. pseudobulking all cell types together) or across major cell types (e.g. all epithelial cells). Then, they can use the single cell data to assess heterogeneity of effect sizes. Another alternative strategy is to apply colocalization analysis to assess if effects are similar or different across celltypes.

8. The eQTLs found in an individual cell-type are perhaps the most susceptible to statistical artifact. Can the authors comment further? Permutation analysis here may be reassuring. Making sure that these genes do not have sparse expression in the cell-type of interest is important. Assessing overlap of SNPs with regulatory structures (Cis-regulatory elements, enhancers, etc) could be informative too.

9. I am a little confused about Figure 4c. Is this a plot of 3,725 genes, or eQTL effects? Appears to be genes, right? Would the pathways and general structure be similar if the authors had simply clustered on gene expression? Why are the IPF implicated genes and the multi-cell eQTLs being analyzed together? I would imagine that the results are mostly driven by the multi-cell eQTLs, and that IPF eQTLs are a relatively small number?

10. I have concerns about the interaction analysis. The number of interacting eQTLs seem high. I also noticed that the authors require only a minimum of 10 samples per group. Finding interactions in such a small number of samples could lead to highly inflated statistics. This may be compounded by the small number of cells creating sparsity. I would ask the reviewers to test using permutations; in this case permuting case-control status would be the way to do it. This way main effects are preserved, and interaction betas should be null. If the p-values are indeed inflated, I would encourage the authors to try using larger cell-type classifications to get around the sparsity issue.

Author Rebuttal to Initial comments

Reviewer #1:

Remarks to the Author:

The authors present the results of single-cell RNA-seq data from lung tissue samples of 116 individuals (67 ILD and 49 unaffected donors) and show that cell specific expression data gives new insights into disease mechanism for ILD than from bulk eQTL data which is an important and novel result.

My main criticism of the paper is that many multi-dimensional patterns in the data are presented in very large and complicated heatmaps with extra information around the peripheries. I don't think these attempts to present "everything" in one plot help the reader get a clear impression of what the data is showing, and personally didn't tell me anything extra to what was said in the text, for example there is just way too much going on in Figure 4b. Could some of the heatmaps be replaced by scatter plots to show trends instead of having to try to compare shades of multiple colours across columns e.g. Figure 4c? Are the numbers feeding into the heatmaps provided in supplementary tables so that if one does happen to spot an interesting difference in shade the numerical values can be obtained?

We want to thank the reviewer for their encouraging comments. We have addressed the reviewer's criticism by (1) simplifying Figure 4 to make it more readable, (2) including additional information in Supplementary Tables 1, 3, and 5, and (3) providing all summary statistics for download.

In general there are too many figures (i.e. the multiple panels in each figure) not all of which are helpful. Figure 1a and 1c should be separate Figures. 1b & 1d could be tables (odd that there are no tables in the manuscript at all).

We have updated **Supplementary Table 1** to include the information presented in **Fig. 1b** and **Supplementary Table 3** with the information presented in **Fig. 1d**. We have elected to keep Fig. 1 panels b and d, as they provide a one-glance overview of the demographic and cell type information associated with the data.

Another concern is the conclusions around the different effect sizes and different distances to TSS sites between sc-eQTL and int-eQTL. The authors mention power effects but I am also wondering is how does the allele frequency spectrum compare between these 2 types of eQTL?

We find a significant difference in minor allele frequencies between sc-eQTL and int-eQTL ($p < 2.2 \times 10^{-16}$), with int-eQTL exhibiting higher MAFs. This difference, however, could also be attributed to power differences, as we have less confidence in int-eQTL with lower MAFs. These results are presented under the subheading “*Disease-specific eQTL are highly cell type-specific.*”

In Figure 4b how exactly were the top-eQTL "pruned" to give a representative sample?

We provide an explanation in the Methods section under “*Assessing significance, sharing, and eQTL classification*”: For each gene, if there is a single top-eQTL, that eQTL is retained. If there are two top-eQTL, the Euclidean distance between the centered absolute values of the estimated effect sizes across cell types for the two eQTL are compared. If the distance is greater than the set threshold (dist=0.2), both are retained. If the distance is less than the threshold then the one that is significant in more cell types is retained. Finally, if there are more than three top-eQTL, the pairwise Euclidean distance between the centered absolute values of the estimated effect sizes for each pair of top-eQTL is calculated. If all pairwise distances are above the threshold, all are retained. Otherwise, hierarchical clustering is performed and the tree is cut using cutree at k between 2 and 5 that maximizes the Silhouette width. For each cluster, the top-eQTL that is significant in most cell types is retained. We have now included a note in this section of the **Methods** that this refers to **Fig. 4**.

I note some marginally significant P-values being presented as supporting conclusions, particularly in Figure 4d (0.043, 0.04 - should be presented as same number of significant figures), is there really an important effect here?

We believe the reviewer is referring to the local false sign rates (lfsr) presented in **Figure 5**. We have used lfsr in place of false discovery rate and considered eQTL with lfsr < 0.05 significant. The lfsr is stricter than the false discovery rate as it requires significant

discoveries to have a consistent sign. A threshold of 0.05 has been previously used in eQTL discovery (see, e.g., 10.1101/2023.05.29.542425).

Also in Discussion "43% of int-eGenes were differentially expressed (adj. p < 0.1)", why such a lenient threshold here, what is the proportion using p < 0.05?

Here, we used a relaxed threshold for differential expression to more reliably select a set of int-eGenes that were *equally* expressed between the two groups. With $p < 0.05$, the number of DE int-eGenes was 32%.

Figure 5b - is a t-test appropriate given the very different shapes of the distributions?

We have repeated this analysis using the non-parametric Wilcoxon sum rank test. These tests yield similarly significant results of $p < 1 \times 10^{-12}$ for each metric.

Difficult to keep track of what is meant by "top-eQTL". Top amongst what? Top for the cell type and gene? Can you be explicit - e.g. most significantly associated eQTL for the tissue/gene?

In the second paragraph of the section titled "*Most eQTL are shared between cell types*", we have defined "top-eQTL" as the most significant eSNP for a given eGene in a given cell type. This definition is used throughout the manuscript.

Supplementary table 1 only has 88 rows and yet it is referred to in a sentence mentioning 116 samples.

This version of **Supplementary Table 1** only contained the information for the previously unpublished sc-RNAseq samples. We have updated this table to include the demographic information for all analyzed samples.

2nd paragraph page 5: "This demonstrated that the relationships between the regulatory mechanisms across lung cell types largely reflected the differences in expression patterns across cell types". Is this as expected? How do you explain the ones that have swapped position such as Mesothelial and CD8/NKT?

Generally, we expected the PCA analyses conducted on the gene expression and eQTLs to share many features. This was driven by the logic that cis-eQTL are drivers of the overall gene expression patterns

observed in a cell type – even with the majority of eQTL being shared across cell types. Conversely, the broader expression program also impacts the cis-eQTL landscape - e.g. genes with high expression levels in a cell type are more likely to be detected as eQTL and transcription factor expression will drive cell type and lineage specific eQTL. Thus, we anticipate a plot very similar to what was presented in the manuscript. Importantly, the PCA loadings were calculated separately on each data set so we anticipate some subtle differences. In the case of Mesothelial cells these occupy a very different space on PC1, but are in similar regions in PC2 in both analyses. Mesothelial cells are a lower abundance cell type with an overall lower number of eQTL which may drive the shift in PC1. CD8 and NK cells do change their relative position but the nearest neighbors remain quite similar.

Also "Top eQTL are considered shared between two cell types if they are significant in both cell types and their mashr estimated effect size is within a factor of 0.5.", what is the rationale for this threshold? How sensitive are results to selection of this threshold?

To our knowledge, there is no universally agreed upon standard for calling an eQTL as shared. Because mashr reduces power imbalance issues and improves estimated effect sizes across cell types, we find that requiring significance and effect size similarity in both cell types is a robust method for calling sharing. The factor approach allows for greater differences in large effect size eQTL and requires small effect eQTL to be more similar, the specific factor was selected because it is easily interpretable and requires eQTL effect sizes to not only be in the same direction but also of a similar magnitude.

2nd paragraph page 8: "Top eQTL were considered to be associated with IPF if the identified eGene was previously reported ($p < 1 \times 10^{-12}$ in an IPF GWAS meta-analysis)". Why this threshold?

We originally chose this threshold to obtain a reasonable number of high-confidence IPF-associated genes (2,406 variants flanking 92 genes) to project this information on the hierarchical clustering of eQTL. This includes the 20 genes reported in Allen et al. 2020, including the 4 genes that replicate across cohorts. We have since revised the figure to simplify it and have removed the annotation for GWAS, as no informative patterns were observed among the clusters presented in **Fig. 4**. We have also provided all the eQTL presented in **Fig. 4** in **Supplementary Table 5**.

Figure 5: DEG should be defined - presumably differentially expression gene?

We have now included a definition of this abbreviation in the figure legend.

Reviewer #2:

Remarks to the Author:

Natri and B Del Azodi et al. analyzed lung tissue single cell RNA sequencing data from IPF and controls and performed eQTL analysis using a pseudo-bulk approach to identify shared and cell type specific eQTLs. Authors describe the patterns of cell type specific eQTLs and identified disease-state interaction eQTLs that show differential regulation pattern by disease status. This study is a valuable resource for lung biology community for providing cell-type specific eQTL data. Pseudo-bulk approach is reasonable, but the study would be even more valuable if true single cell level heterogeneity could be incorporated in the analysis - such as eQTLs of cellular trajectories associated with alveolar epithelial regeneration and repair in this dataset with pulmonary fibrosis.

Major comments

1. Fig 4b: I believe the purpose of the figure is to demonstrate multi-cell type eQTLs tend to be lineage specific with similar effect sizes by lineage, but too much information is displayed which makes it less intuitive. Consider adding labels to the cluster figure as the text describes clusters by number, and focusing on a few representative, disease relevant eQTLs (that show concordant and discordant effects by cell types for example) in the main figure may be more helpful.

We thank the reviewer for this comment which was also brought up by reviewer 1. We have revised **Fig. 4** by moving the eQTL metric violin plots to **Figure S9** and by simplifying the heatmap annotations.

2. Disease-specific eQTL, such as DSP is very interesting that same locus has differential regulatory effects. The authors examined TF motifs and differential expression of TFs but it is not directly linked back to DSP. Was TFs differentially expressed among the different genotype groups by disease status? Was this variant associated with DSP or other IPF risk variants?

The int-eQTL SNP rs2003916 was not significantly associated with IPF risk in the meta-analysis ($p=0.03377$). When contrasting donors with 0/0 genotypes for rs2003916 and those with at least one ALT allele or those with two ALT alleles, we find no differential expression (thresholds for DE: $AUC>0.6$ and

adj. $p < 0.01$ with Wilcoxon auROC method in *presto*) of the significantly enriched TFs in any of the epithelial cell types included in the eQTL analysis.

3. MUC5B GWAS variant colocalization is significant in AT2, transitional AT2, secretory cells and macrophages while MUC5B is predominantly expressed in one secretory cell type (SCGB1A1+MUC5B+). As SCGB1A1+ secretory cells can transdifferentiate into other epithelial cell types in IPF, incorporating trajectory analysis and examining the dynamic eQTL pattern would further clarify the causal cell type.

We agree that is interesting, but pseudobulk-based eQTL quantification inherently aggregates data from all of a given cell-type, so the effect cannot be modeled across a trajectory – that would require a fundamentally different approach - treating each individual cell as an observation. While we are excited about the concept of such analyses, we are not aware of sufficiently developed rigorous methods that are currently available to make such an analysis robust. Indeed our group is working to develop methods in this space, but they are not yet ready to be deployed in this manuscript.

4. For single-cell type eQTLs, how many of them were explained by cell-type specific expression vs. more globally expressed but showing specific regulatory effect on a single cell type?

Cell type-specific eQTL were detected for 29 cell types, and the 2,332 eQTL that were specific to a single cell type were associated with 1,828 genes. Out of these genes, 1,411 had eQTL only in one cell type. Out of these, 584 genes met the threshold of >5% of cells expressing the gene with a log₂-transformed read count of 1. These eGenes were detected across 24 cell types. Out of these 584 robustly expressed cell type-specific eGenes, most met the expression threshold in multiple cell types. 13 were only expressed in the given cell type, and 14 were globally expressed across all cell types. As the genes included in eQTL testing were selected based on a threshold across all cell types, genes with lower expression levels were included in the analysis for each cell type. We have added a new **Supplementary Figure 10** to present the expression levels of these eGenes to demonstrate that the cell type-specific eQTL are not largely driven by cell type-specific gene expression.

Supplementary Figure 10. Expression of eGenes unique to a single cell type (n=584).

Of the int-eGenes, 43% were differentially expressed (adj. $p < 0.1$) between ILD and unaffected samples in the particular cell type.

5. KRT5-/KRT17+ cells are excluded from disease interaction analysis as it probably is not expressed enough in the non-diseased sample, however as it is a unique pathologic cell type in fibrosis, would be interesting to show the eQTL result/discussion on its findings. Similar to comment on 3 –trajectory analysis with AT2 cells and KRT5-/KRT17 will highlight the utility of single cell RNAseq.

We thank the reviewer for this comment. We obviously agree that KRT5-/KRT17+ cells are interesting! These cells were excluded from the interaction analysis due to insufficient cells in the control samples. Part of our motivation to build new eQTL models that capture the plasticity of cell types was due to our observations of the AT2 -> KRT5-/KRT17+ transitions. As we mentioned above, these models are not yet fully developed but we hope to apply them to this trajectory in the future.

Minor comments:

1. Providing the number of cells and relative abundance of cell types per subject and median number of genes in table S1 would be helpful.

We have now included the number of cells, the relative abundance of cell types, and the proportion of genes included in the eQTL analysis that were expressed in at least 1%, 5%, or 10% of cells in at least one of the analyzed cell types for each sample in **Supplementary Table S1**.

2. Was regional heterogeneity incorporated in the analysis?

Our samples include paired more and less fibrotic samples from donors with ILD. In the pseudobulk-eQTL analysis, we have aggregated gene expression across all cells from each donor, thus aggregating information across more and less fibrotic regions. We agree that it would be interesting to explore relationships within a given sample, but this dataset lacks sufficient statistical power for these analyses. We hope to perform these in the future on a larger dataset sufficiently powered for such an approach.

3. What covariates were used for the LIMIX? Smoking status will confound SNP-gene expression relationship, was it adjusted?

As described in the **Methods**, in order to control for unwanted technical effects as well as known and unknown confounders, we regressed out the first 20 cell type expression principal components before model fitting. We also accounted for variance due to population structure by including a random effect in the linear mixed model with covariance defined by an identity-by-descent relationship matrix between individuals. Further, to account for differences in cell type abundance across donors, we

included the number of cells aggregated as a second random effect. Random effects were marginalized from the model using the low-rank optimization method. We find that smoking status and other known covariates are correlated with PCs accounted for in our analysis, for example, PC2 accounts for tobacco usage in the epithelial AT2 model (new **Supplementary Figure 5**).

Fig. S5: Heatmap of correlations between the PCs included as covariates in the eQTL analysis of AT2 cells and known covariates.

Reviewer #3:

Remarks to the Author:

Natri et al presents a large data set of lung single cell data, and map eQTLs in this single cell cohort. The work is timely and to my knowledge reflects the first single cell eQTL study in lung. However, I think there are a lot of technical concerns that I have. Single cell data can be sparse, and pseudobulk profiles may have variable sparsity if different numbers of cells are used to create profiles. I am worried that they may have a large number of false positives and may have inflated statistics. My major comments are below:

1. I think it would be useful to assess how well their results co-localize with GTEx lung. I think this would be a useful sanity check for their results, and would give a sense as to what is being missed in bulk analyses. Currently there is some mention of overlap in Figure 5f, but this may be confounded by thresholding. Specifically, I would like to know, for lead SNP-gene pairs out of their data, how well do effect sizes (betas) correspond to GTEx betas for the same SNP-gene pair. Also – for a given tissue, what percentage of discovered eGenes colocalize with lung GTEx results. Is GTEx capturing some cell-types better than others?

As per the reviewer’s helpful suggestion, we have extended our analysis of the replication of sc-eQTL among GTEx lung eQTL in **Supplementary Note 1** and **Supplementary Figures S14** and **S15**. Overall, we find a significant correlation between cell type-eQTL and GTEx bulk-eQTL across cell types and tissues. This correlation is the strongest between immune cell type sc-eQTL and GTEx whole blood and non-immune sc-eQTL and GTEx lung. These patterns are also observed in the colocalization analysis.

Fig. S14: Correlation of eQTL effect sizes (R^2) between cell type-eQTL and GTEx bulk eQTL (above), and for each cell type, the proportion of the tested genes that colocalized with GTEx bulke-eQTL (below).

Fig. S15: eQTL effect sizes from GTEx lung, whole blood, and brain cortex (y-axis) and epithelial AT2 cells and monocyte-derived macrophages (x-axis).

2.1 had a few questions about the single cell analysis. The authors used shared—nearest neighbor strategy to perform batch correction. Given the tissue derived nature of their data set I would worry about batch effects. I could not find any discussion around the potential for batch effects in their data. Can the authors provide some metrics and reassurance that batch has been adequately addressed? Given the relatively small number of cells per cell-type per individual, batch effects may be a particular issue.

We have conducted additional analyses to evaluate batch mixing in our scRNAseq data, presented in a new **Supplementary Figure 2**. We also note that in our eQTL analysis, we control for any remaining batch effects by including the top 20 PCs as covariates in the LMM (new **Supplementary Figure 5**).

3. I noticed that the authors did build pseudobulk profiles from cell-types in individuals with as few as 5 cells. That strikes me as somewhat dangerous, given how sparse single cell data can be (as few as 1000 non-zero genes per their QC). I would expect that in these pseudobulk profiles that many zero count genes are present. I think the authors probably need to apply more stringent QC to their pseudobulk profiles. Given the number of cell types examined (43), and the number of individuals (100) assayed, and the cells generated (500K), it means that on average pseudobulk profile is only constituted from ~100 cells. They should assess the number of zero count values for genes in their analysis. If a profile has too many zero count values, probably it should be removed. If there is too much sparsity in their data, authors may want to consider using coarser cell-type definitions.

We agree with the reviewer that sparsity is a difficult issue to address in analyses such as this. We used 5 cells as a minimum threshold to include the donor in that cell-type analysis as recommended by Cuomo, Alvari, Azodi et al. 2021. This threshold was recommended to minimize donor loss (which would have a substantial negative impact on power) while removing donors with limited expression information. They show that using a mean aggregation approach, this threshold was effective at generating pseudobulk counts that replicated eQTL from bulk tissue studies. In our work, to further address the issue of sparsity, we applied a filtering to genes to only include genes in our analysis which had sufficient levels of expression by requiring both breadth (expression in >10% of cells) and magnitude (mean counts > 0.1 across all cells) of expression. With these QC filters, the median frequency of zeros across genes and celltypes is just 9% (see new Supplemental Table 4). Finally, because mashr requires a complete eQTL matrix (i.e., no missing estimates for any eQTL for any celltype), we do run limix eQTL mapping on some genes for some celltypes where there is poor power. However, after applying mashr, we apply a stricter inclusion filter where mashr-adjusted eQTL effects are only reported for genes meeting the expression criteria for that given celltype.

4. Author's should use a computational strategy like scrublet to remove doublets. Currently it appears this is done qualitatively.

We have assessed numerous methods to identify and remove doublets, including tools such as DoubletFinder, which has been reported to exceed scrublet in accuracy (10.1016/j.cels.2020.11.008). We find that our current approach performs equally well or better than strictly computational approaches: in the epithelial cell population, DoubletFinder only recovers 8,230 doublets (3.7%), while our approach identifies 18,588 doublets (8.5%), which is more in line with our expectation. Thus, have elected to employ this strategy.

5. It isn't clear to me what type of SNP QC was done by the authors. Typically low quality imputed SNPs are removed.

Biallelic, autosomal SNPs were filtered to include SNPs with a minor allele frequency > 5%, Hardy-Weinberg equilibrium $p > 1 \times 10^{-6}$, and further pruned to remove highly correlated SNPs, resulting in ~1.9 million SNPs. We have now included additional permutation analyses to demonstrate the robustness of our findings.

6. Single cell eQTL maps are susceptible to statistical inflation. Given the relatively small number of individuals in their study (~100, with individual cell types having as few as ~50), and the potential for sparsity or outlier expression values to not work well with linear models, there is the potential for inflated p-values. To assess the possibility, I think it is important for authors to do permutations. The appropriate permutations here would be to reassign whole-genome genotype data to single cell data. E.g. each individual gets someone else's genome-wide genotype data. They should then carry through their statistical procedure. If robust, they should find very few if any eQTLs. While cumbersome – it is the only way to understand if they are producing false positives in their analysis.

We thank the reviewer for this helpful suggestion. We employed a permutation analysis by shuffling genotypes and repeating the eQTL and mashr analyses for each cell type and comparing the observed p -values to permuted p -values (new **Supplementary Figure 6**). Further, we compared the permuted p -values to the expected statistical p -values under the null hypothesis (new **Supplementary Figure 7**). We observe no notable deviation of the permuted p -values compared to the null distribution, demonstrating that our approach is well-calibrated to avoid false positives.

Fig. S6: Quantile-quantile plots for each cell-type showing the observed empirical p-values of the top hit per gene (y-axis) against the permutation-based (genotypes were shuffled independently for each cell-type) empirical p-values of the top hit per gene (x-axis). Empirical p-values are from the limix sc-eQTL mapping runs and are shown on the $-\log_{10}$ scale. Observed and permuted values were sorted from largest to smallest.

Fig. S7: QQ plots for each cell-type showing the expected statistical p -values under the null hypothesis (x-axis) against the permutation-based statistical p -values using limix to run sc-eQTL mapping with permuted genotypes (y-axis). The null hypothesis was generated by, for each gene, taking the minimum value after sampling N from a uniform distribution (min=0; max=1), where N is the number of SNPs tested for that gene.

7. To identify eQTL sharing, the authors identify the top SNP-gene pair for each cell-type and then assess consistently across cell types. This approach might suffer from thresholding of effects. One alternative strategy that might be more accurate might be to map eQTLs across all celltypes (e.g. pseudobulking all

cell types together) or across major cell types (e.g. all epithelial cells). Then, they can use the single cell data to assess heterogeneity of effect sizes. Another alternative strategy is to apply colocalization analysis to assess if effects are similar or different across celltypes.

To reduce the effect that thresholding would have on our eQTL sharing analysis, we (1) applied mashr, (2) used a two-stage significance thresholding that used a less stringent threshold for an eQTL in any second cell type to be considered significant, and (3) considered eQTL to be shared if their mashr estimated effect sizes were within a given factor.

8. The eQTLs found in an individual cell-type are perhaps the most susceptible to statistical artifact. Can the authors comment further? Permutation analysis here may be reassuring. Making sure that these genes do not have sparse expression in the cell-type of interest is important. Assessing overlap of SNPs with regulatory structures (Cis-regulatory elements, enhancers, etc) could be informative too.

In addition to the permutation analyses and inspecting the sparseness of our expression data, we have overlapped our different classes of cell type-eQTL as well as the int-eQTL with genic annotations from TxDb. These analyses reveal no statistically significant differences between the different types of eQTLs when inspecting the proportions of eQTL along promoters, genic, and intergenic regions (**Supplementary Figure S12**). We now present these results in the sections “*Most eQTL are shared between cell types.*” In the absence of reliable annotations for the relevant cell types, we did not pursue analyses on enhancers.

9. I am a little confused about Figure 4c. Is this a plot of 3,725 genes, or eQTL effects? Appears to be genes, right? Would the pathways and general structure be similar if the authors had simply clustered on gene expression? Why are the IPF implicated genes and the multi-cell eQTLs being analyzed together? I would imagine that the results are mostly driven by the multi-cell eQTLs, and that IPF eQTLs are a relatively small number?

Figure 4 is of eQTL effect sizes. We have revised the figure so that eQTL effect sizes are shown in gray for genes expressed in <10% of cells of that cell-type. We included the IPF GWAS variants to explore whether they cluster with specific classes of multi-cell type-eQTL. However, we observe no significant enrichment of GWAS SNPs among the clusters in the old Fig. 4c. We have simplified the figure to clarify its purpose, which is to demonstrate the high level of lineage sharing of multi-cell type-eQTL, by

excluding some of the detailed annotations. We have also shared the eQTL presented in this figure in **Supplementary Table 5**.

10. I have concerns about the interaction analysis. The number of interacting eQTLs seem high. I also noticed that the authors require only a minimum of 10 samples per group. Finding interactions in such a small number of samples could lead to highly inflated statistics. This may be compounded by the small number of cells creating sparsity. I would ask the reviewers to test using permutations; in this case permuting case-control status would be the way to do it. This way main effects are preserved, and interaction betas should be null. If the p-values are indeed inflated, I would encourage the authors to try using larger cell-type classifications to get around the sparsity issue.

In our int-eQTL analysis, after running mashr we detect 83,596 int-eQTL. Following the reviewers suggestion, we permuted the disease status of the individuals and then repeated this analysis (interaction-eQTL mapping with limix + mashr + calling significance with lfsrs) and using the same significance thresholds only 829 int-eQTL from the permuted run were considered significant. These results support a 1% false positive rate in our int-eQTL mashr results, demonstrating that our minimum requirements for inclusion in the interaction eQTL mapping study were sufficient. We now report these results in the manuscript under *"Disease-specific eQTL are highly cell type-specific."*

Decision Letter, first revision:

7th Nov 2023

Dear Professor Banovich,

Your Article, "Cell type-specific and disease-associated eQTL in the human lung" has now been seen by 3 referees. You will see from their comments below that while they find your work of interest, some important points are raised. We are interested in the possibility of publishing your study in Nature Genetics, but would like to consider your response to these concerns in the form of a revised manuscript before we make a final decision on publication.

To guide the scope of the revisions, the editors discuss the referee reports in detail within the team, including with the chief editor, with a view to identifying key priorities that should be addressed in revision and sometimes overruling referee requests that are deemed beyond the scope of the current study. In this case, we would like you to address Reviewers' comments in full. We hope that you will find the prioritized set of referee points to be useful when revising your study. Please do not hesitate to get in touch if you would like to discuss these issues further.

We therefore invite you to revise your manuscript taking into account all reviewer and editor comments. Please highlight all changes in the manuscript text file. At this stage we will need you to upload a copy of the manuscript in MS Word .docx or similar editable format.

*2) If you have not done so already please begin to revise your manuscript so that it conforms to our Article format instructions, available [here](http://www.nature.com/ng/authors/article_types/index.html). Refer also to any guidelines provided in this letter.

[redacted]

We hope to receive your revised manuscript within four to eight weeks. If you cannot send it within this time, please let us know.

Nature Genetics is committed to improving transparency in authorship. As part of our efforts in this direction, we are now requesting that all authors identified as 'corresponding author' on published papers create and link their Open Researcher and Contributor Identifier (ORCID) with their account on the Manuscript Tracking System (MTS), prior to acceptance. ORCID helps the scientific community achieve unambiguous attribution of all scholarly contributions. You can create and link your ORCID

from the home page of the MTS by clicking on 'Modify my Springer Nature account'. For more information please visit www.springernature.com/orcid.

Sincerely,
Chiara

Chiara Anania, PhD
Associate Editor
Nature Genetics
<https://orcid.org/0000-0003-1549-4157>

Referee expertise:

Referee #1:

Referee #2:

Referee #3:

Reviewers' Comments:

Reviewer #1:

Remarks to the Author:

Figure 4 is clearer and more concise now.

Even though the authors have added the category labels for diagnosis, smoking status and ethnicity to Supplementary Table 1 (although I think they were there in the previous version) it is still not possible to obtain the actual percentages of each category easily from Figure 1b (nor is this summary given in Supplementary Table 1) due to the inherent problem with stacked bar charts i.e. you can only read directly off the Y-axis scale what the percentage of the lower category is, also the choice of Y-axis tick marks is poor in that we have ticks at 0, 12.5, 25, 37.5 etc. making extracting the information at a glance even harder. Can you add the actual percentages below each category label in Figure 1b e.g.

Control
(37.8%)

And perhaps add the summary percentages to Supplementary Table 1.

In Figure 5d I wasn't questioning the choice of $p < 0.05$, just that the results only just passing this threshold aren't as convincing, especially as 4 SNPs are presented, unless the p is also adjusted for the number of cell types tested? In the statement: "including rs2003916, which was not significantly associated with IPF risk in the GWAS meta-analysis ($p=0.03377$)...", I think the standard error has

been given instead of the P value from the GWAS meta-analysis. However, I do not agree with the statement as it stands as a single SNP lookup does not require a correction for multiple testing. The same number of significant digits should be displayed for the *Isfr* for *Sec*, *SCGBA32+*.

In Discussion "43% of int-eGenes were differentially expressed (adj. $p < 0.1$)", the authors say "Here, we used a relaxed threshold for differential expression to more reliably select a set of int-eGenes that were equally expressed between the two groups", this reason for this more lenient threshold should be stated in the text.

I am satisfied with the responses and explanations to all my other outstanding comments and queries

Reviewer #2:

Remarks to the Author:

Authors have addressed most of the points I raised, except the following minor points.

Fig 4 – in the revised figure authors omit the cell types previously specified in the bottom of the heatmap which makes the figure incomplete to understand, especially as the supplementary file is just listing the variants without cell types and effect sizes. The supplementary table should be expanded to include cell types and effect sizes.

Reviewer #3:

Remarks to the Author:

Responses to my comments. (Reviewer #3)

Authors have addressed some of my comments. I will say that their responses were somewhat more terse than I am used to, and in some instances I felt that they could have addressed comments better.

Comment 1

The authors examined colocalization between GTEX and their eQTLs. I felt that they did a minimal analysis here. Notably the colocalization seems pretty low. Do the authors have any comment on this? How well do their loci localize with non-lung, and non-blood tissues?

Comment 2

The authors did not really make any effort to evaluate their batch correction strategy rigorously, or compare it to other batch correction strategies. Taken at face-value, Supplementary Figure S2 reveals substantial batch effects.

Comment 3

I expressed concerns about the small number of cells in pseudobulk profiles. It isn't clear from the author's response, that they investigated the potential for the small number of cells for some profiles to be an issue or not. They cited Cuomo et al to defend their choice. However, few single cell eQTL studies have been done, and I am not sure that there is a good understanding of what an appropriate threshold is here.

Comment 4.

I asked the authors what strategy they used to remove Doublets. In the response letter they indicated they used Doubletfinder. However, it doesn't appear that this is described (or justified) in the methods of the main text. This is an important choice and should be described.

Comment 8

The authors did not bother to assess enrichment of their eQTLs in regulatory elements at all.

Author Rebuttal, first revision:

Reviewers' Comments:

Reviewer #1:

Remarks to the Author:

Figure 4 is clearer and more concise now.

Even though the authors have added the category labels for diagnosis, smoking status and ethnicity to Supplementary Table 1 (although I think they were there in the previous version) it is still not possible to obtain the actual percentages of each category easily from Figure 1b (nor is this summary given in Supplementary Table 1) due to the inherent problem with stacked bar charts i.e. you can only read directly off the Y-axis scale what the percentage of the lower category is, also the choice of Y-axis tick marks is poor in that we have ticks at 0, 12.5, 25, 37.5 etc. making extracting the information at a glance even harder. Can you add the actual percentages below each category label in Figure 1b e.g.

Contr

ol

(37.8

%)

And perhaps add the summary percentages to Supplementary Table 1.

We thank the Reviewer for their comments, and we are glad that our improvements were well received. We appreciate that it was still difficult to obtain actual percentages from these plots/tables and we have now added these percentages directly to the

figure legend under Figure 1b: “ Percentage proportions of donors by diagnosis (42.1% unaffected control, 34.2% IPF, 23.7% other ILD), self-reported ethnicity (66.7% European, 9.6% African American, 17.5% N/A, 6.1% other), and smoking history (46.5% ever smoker, 29.8% never smoker, 23.7% N/A).”We ultimately did not add these percentages to Table S1 as this table is on a per individual basis.

In Figure5d I wasn't questioning the choice of $l_{sfr} < 0.05$, just that the results only just passing this threshold aren't as convincing, especially as 4 l_{sfr} s are presented, unless the l_{sfr} is also adjusted for the number of cell types tested?

We thank the Reviewer for clarifying their question and further inquiring on the significance of the result presented in the figure. We are utilizing mashR for multivariate adaptive shrinkage; instead of relying on condition-by-condition effect size measurements and significance levels, this approach leverages information across all tested conditions, in this case, cell types. The l_{sfr} , which is analogous to FDR, is more stringent as it considers the direction of the effect across conditions (that is, low l_{sfr} indicates high confidence in the sign of an effect across our cell types). As this method and metric are optimized for the analysis of effect sizes across multiple conditions, further adjusting for multiple cell types tested is not necessary. A similar approach and threshold have been used in, for example, the analysis of GTEx eQTL across multiple tissues (doi:10.1126/science.aaz1776).

In the statement: “including rs2003916, which was not significantly associated with IPF risk in the GWAS meta-analysis ($p=0.03377$).....”, I think the standard error has been given instead of the P value from the GWAS meta-analysis. However, I do not agree with the statement as it stands as a single SNP lookup does not require a correction for multiple testing.

We thank the Reviewer for spotting this error. We have revised the sentence as follows: “ including rs2003916, which was not significantly associated with IPF risk in the GWAS meta-analysis ($p=0.15$).” As this variant is not associated with the trait even at a nominal p-value threshold, we believe our statement that this variant was not associated with IPF risk accurately reflects available data.

The same number of significant digits should be displayed for the l_{sfr} for Sec, SCGBA32+. In Discussion "43% of int-eGenes were differentially expressed (adj. $p < 0.1$)", the authors say "Here, we used a relaxed threshold for differential expression to more reliably select a set of int-eGenes

that were equally expressed between the two groups”, this reason for this more lenient threshold should be stated in the text.

We thank the Reviewer for noting the differences in digits on the figure and for inquiring about the differential expression analysis. We have revised the figure to include the same number of digits for lfsr in Figure 5d to address the Reviewer's concern. We have removed the use of the word “relaxed” threshold from this paragraph, as an FDR of 10% is a generally accepted cut off for multiple testing of DE genes. In other studies we have used more stringent cutoffs, thus the use of the term “relaxed”, but after your comment we decided it would be more appropriate to change the text on page 11 to the following: “Out of the 37 genes encoding TFs disrupted by int-eQTL that were also tested for differential expression, 30 were DE between ILD and unaffected samples in at least one cell type when employing a significance threshold of adj. $p < 0.1$.” Furthermore, we have clarified in the text that we considered genes to be equally expressed between cases and controls with an adj. $p > 0.1$.

I am satisfied with the responses and explanations to all my other outstanding comments and queries

Reviewer #2:

Remarks to the Author:

Authors have addressed most of the points I raised, except the following minor points.

Fig 4 – in the revised figure authors omit the cell types previously specified in the bottom of the heatmap which makes the figure incomplete to understand, especially as the supplementary file is just listing the variants without cell types and effect sizes. The supplementary table should be expanded to include cell types and effect sizes.

We thank the Reviewer for their continued feedback with regards to Figure 4. We agree that the current figure would be improved by a full table with cell types and effect sizes. We have updated Table S5 to include the mashR posterior effect size estimates for each cell type.

Reviewer #3:

Remarks to the Author:

Responses to my comments. (Reviewer #3)

Authors have addressed some of my comments. I will say that their responses were somewhat more terse than I am used to, and in some instances I felt that they could have addressed comments better.

We sincerely thank the Reviewer for their thoughtful engagement in this work and for the critical comments that have enabled us to strengthen this manuscript. We strive to be direct and succinct in our responses and regret that our attempts to be concise resulted in responses that came across as terse. We are grateful for this opportunity to more fully address the Reviewer's comments.

Comment 1

The authors examined colocalization between GTEx and their eQTLs. I felt that they did a minimal analysis here. Notably the colocalization seems pretty low. Do the authors have any comment on this? How well do their loci localize with non-lung, and non-blood tissues?

We thank the Reviewer for further inquiring about the level of replication of our cell-type eQTL among bulk tissue eQTL. We have now expanded this analysis and, additionally, compared our findings with previously published cell type-eQTL studies. These expanded findings are contained within Supplementary Note 2 and Figure S15 as well as in the main text on page 11 and 12.

In our comparison with GTEx, we have included lung, whole blood, as well as brain cortex bulk-eQTL as a non-blood tissue. To our knowledge, previous cell type-eQTL studies have not used a colocalization analysis to evaluate the level of replication among GTEx or other bulk tissue eQTL, making it impossible to compare our results to other work. However, we do find our colocalization analysis captures cell lineage level effects – specifically immune cell eQTL tend to have higher colocalization with GTEx whole blood, while non-immune cells tend to have higher colocalization with GTEx lung.

In our additional analysis, we found the effect sizes of cell type and bulk-eQTL to be highly correlated with an R^2 of 0.318 between AT2 cells and GTEx lung. In contrast, comparisons between lower abundance cell types and non-lung bulk-eQTL resulted in the lowest correlations, with an R^2 of 0.0883 between cDC1 cells and brain cortex. Finally, we have to explore the level of top-eQTL overlap with GTEx on a cell-type

level, employing the same significance threshold as in our enrichment analysis (GTEx nominal $p < 1 \times 10^{-6}$). We find that up to more than 10% of cell type-eQTL replicate in GTEx, with the highest overlap (10.37%) between Inflammatory monocytes and whole blood eQTL and the lowest (0.79%) between Alveolar fibroblasts and brain cortex. All three approaches, and the colocalization and effect size correlation, in particular, reveal a lineage-specific pattern of overlap that reflects the expected similarity of cell types and tissues included in the analysis.

To contextualize our results, we compared the replication of cell type eQTL found in our study to a previous study of eQTL among 6 PBMC cell types (Yazar et al. 2022, n=982, 10.1126/science.abf3041). In this study, the authors found, 40.4% of cell type-eQTL replicated among GTEx whole blood. When employing the same significance threshold as Yazar et al., we find that 12.6% of our immune cell type-eQTL replicate among GTEx blood, and 11.6% of all cell type-eQTL in our analysis replicate in GTEx lung. Given the smaller sample size in our study (114 vs 982) and the complexity of the lung compared to blood a modest reduction in replication is expected. Further, using the threshold employed by Yazar et al., 36.3%, 28.5%, and 38.3% of the GTEx lung, whole blood, and brain cortex eQTL were significant eQTL in at least one of the cell types in our study. We further compared the eQTL detected by Yazar et al. to our cell type-eQTL. Out of the 848 eQTL for NK cells and 104 eQTL for plasma cells detected by Yazar et al. that were also tested for in our study, 31.0% and 19.2% were significant in our analysis of these cell types, respectively.

Fig. S15: For each cell type, the proportion of the tested genes that colocalized with GTEx bulk-eQTL (top), the correlation of eQTL effect sizes (R^2) between cell type-eQTL and GTEx bulk eQTL (middle), and the proportion of eQTL that replicate in GTEx with a nominal $p < 1 \times 10^{-6}$ (bottom).

Comment 2

The authors did not really make any effort to evaluate their batch correction strategy rigorously, or compare it to other batch correction strategies. Taken at face-value, Supplementary Figure S2 reveals substantial batch effects.

We thank the Reviewer for further scrutinizing our integration approach. We have now expanded our description and analyses on batch correction strategies related to both our single cell data integration as well as our eQTL analysis.

Beginning with data integration, we have compared three different methods using the epithelial subset of the dataset: rPCA, which we have employed in the current study, as well as Harmony and “atomic-sketch” integration. We find that all three methods perform similarly, with Harmony integration resulting in the highest level of batch mixing, rPCA performing next best, and atomic sketch performing the worst (iLISI 5.55, 5.46, and 5.22 respectively). However, we found Harmony tended to mute some biological signal possibly through overintegration, resulting in less clear separation between cell types – a feature noted by some benchmarking papers (<https://www.nature.com/articles/s41467-023-37126-3>, <https://www.biorxiv.org/content/10.1101/2021.08.04.453579v1.full>). Given the similarities in iLISI score between rPCA and Harmony, and weighing the trade-off of batch mixing and preservation of biological variation, we have decided to employ rPCA, which performs well on large datasets. We have incorporated these comparisons into Supplementary Figure 1.

Furthermore, we have added a new Supplementary Figure 2 of Pearson residuals when comparing the observed proportions of cell types across batches to the expected proportions. While in some cases, the proportions deviate from the expected, we note that these deviations are primarily observed in cell types/batches with fewer cells (Fig Sx).

Finally, with respect to the eQTL models, we account for batch effects by regressing out the effects of the first 20 expression PCs. These PCs capture sequencing batches (Fig. S6, previously S5). This approach substantially increases power in eQTL studies ([10.1371/journal.pcbi.1000770](https://doi.org/10.1371/journal.pcbi.1000770)) and has been widely used in bulk-eQTL studies ([0.1126/science.aaz1776](https://doi.org/10.1126/science.aaz1776)) as well as pseudobulk cell type-eQTL studies ([10.1038/s41593-022-01128-z](https://doi.org/10.1038/s41593-022-01128-z), [10.1126/science.abf3041](https://doi.org/10.1126/science.abf3041)).

Fig. S1: a, UMAP dimensionality reductions of cells included in the pseudobulking and eQTL mapping, pseudocolored by flowcell, processing site (TGen or Vanderbilt), cell cycle phase, proportions of mitochondrial reads, number of read counts, and number of features. **b**, Comparison of three

integration methods across the epithelial cell types. c, iLISI for batch mixing with the three integration methods.

New Fig. S2. χ^2 residuals of the observed and expected proportions of cell types across batches.

Fig. S6: Heatmap of correlations between the PCs included as covariates in the eQTL analysis of AT2 cells and known covariates.

Comment 3

I expressed concerns about the small number of cells in pseudobulk profiles. It isn't clear from the author's response, that they investigated the potential for the small number of cells for some profiles to be an issue or not. They cited Cuomo et al to defend their choice. However, few single cell eQTL studies have been done, and I am not sure that there is a good understanding of what an appropriate threshold is here.

We thank the Reviewer for probing deeper about the potential effects of the choice of

lower threshold for the number of cells per cell type for pseudobulk profiles. As we believe our expanded answer will be of value to others, a slightly modified version of the response below has been added as Supplementary Note 1.

A current challenge in the field, as the Reviewer identifies, is that few single-cell eQTL studies have been done, so as a field we lack the empirical evidence that would otherwise inform rules of thumb or heuristics like these. Eventually, as we gain

experience we will likely see convergence on standards like we have with, for example, minimum sample sizes for eQTL mapping per tissue in GTEx. In the meantime, however, we have experience from only a small number of empirical studies to draw on and those studies, like ours here, do not have “ground truth” available to us to determine whether or not a particular choice of cell-number-threshold is optimal.

Facing this situation, the best option we have for setting a cell-number-threshold is to draw on the detailed investigations that Cuomo et al undertook to define current “best practices” for single-cell eQTL mapping. It is unlikely that the Reviewer missed this detail, but for full transparency DJ McCarthy was a senior author on the Cuomo et al paper - as such we know it well. We cite that paper not only to defend our choice, but because it is the best source of information for an evidence-based choice of threshold given the current state of empirical experience in the sc-eQTL field. Although not a headline focus of that paper, the authors comment on the choice of 5 cells as a minimum threshold for inclusion of a pseudobulk profile:

“In all cases (i.e., using any of the aggregation methods), aggregated expression values were only calculated for samples (i.e., donors or donor-run combinations) with at least 5 cells. This threshold was selected to be loose enough to minimize donor loss, while still eliminating donors with poor expression support.” [early in the results section]

Although brief, this comment captures the dilemma in setting a cell-number-threshold for pseudobulking. There is always a tradeoff:

Option	Benefit	Downside
---------------	----------------	-----------------

Higher cell number threshold per pseudobulk profile	Less noisy pseudobulk profiles as they average over more single cells	Loss of cell types for analysis (e.g., if they have too few donors for inclusion, say <70 donors); Loss of power in remaining cell types due to loss of donors/individuals and thus reduced sample size for eQTL mapping in some cell types
Lower cell number threshold per pseudobulk profile	Retain more cell types for analysis, getting a fuller picture of genetic regulation of gene expression throughout the tissue; Maximize eQTL detection power	Noisier pseudobulk expression profiles

On balance, we found in that paper that the better balance was to err on the side of a lower cell-number-threshold to maximize power increases through maximizing the number of donors (and in our setting cell types too). Detailed simulation results that precisely define the tradeoffs of different choices of cell-number-threshold were not shown in that paper. However, the overall results shown in that paper that support the use of a cell-number-threshold of 5, as small as that may seem, as this threshold was used for the simulation studies and detailed benchmarking of single-cell eQTL mapping results against those derived from matched bulk RNA-seq data.

In particular, the results in Fig S7 from Cuomo et al on effects on power, empirical FDR, and beta correlation of the number of donors and the average number of cells per donor are relevant for this discussion. The caption for Cuomo Fig S7 provides information on the distributions used to allocate cell numbers to individuals for the simulations underpinning those plots. The R code showing the distributions of cell number values per donor for an average of 50 cells per donor and an average of 120 cells per donor are shown below:

```
> ## settings from Cuomo et al "best practices" paper for ave 50 cells per donor and ave 120 cells per donor
> set.seed(101)
> ns <- round(sort(rgamma(50, shape = 2.1, rate=0.04)))
> summary(ns)
```

```

Min. 1st Qu. Median Mean 3rd Qu. Max.
8.00 24.25 44.00 48.68 67.00 157.00

> ns <- round(sort(rgamma(50, shape = 2.1, rate=0.017)))
> summary(ns)
Min. 1st Qu. Median Mean 3rd Qu. Max.
23.00 59.25 113.50 123.46 157.75 481.00

```

First, we can see that the simulations include a distribution of cell numbers per donor, down to small (i.e., single-digit) numbers of cells for some donors in each simulation run, especially for the setting for an average of 50 cells per donor. The range of cells-per-donor explored in the simulations closely matches the empirical data in our studies, as we can see from Supplementary Table S3 in this manuscript:

```

> summary(y$`Mean Cells Per Donor`)
Min. 1st Qu. Median Mean 3rd Qu. Max.
13.16 48.62 80.98 127.11 122.98 801.57

```

So around 75% of the 38 cell types we study have at least 50 cells per donor, on average, and the middle 50% of cell types in our study have an average number of cells per donor between 50 and 120. Thus, the simulation settings explored in the Cuomo et al paper are directly relevant to our study here. Looking closely at Fig S7b (and focusing on the “dr-mean” results in dark blue as they match our approach in this study), we see very consistent values for the empirical FDR across values for average # cells per donor (on average the empirical FDR at around 0.07 is a little higher than the nominal FDR of 0.05 whatever the average number of cells per donor used). As such, the eQTL mapping results are not enriched for false positive results if we allow for a smaller number of average cells per donor. (There is a strong effect on power for average # cells per donor - obviously if we have more cells available for analysis we should use them!). In Fig S7a,

however, we do see an elevated empirical FDR for 50 donors relative to 87 donors and more. Thus, we should endeavor to maximize the number of donors available for analysis to bring the empirical FDR as close as possible to the nominal FDR. We can also see a stark reduction in eQTL discovery power for 50 donors compared with more donors.

Across the 38 cell types we study for eQTL mapping in this work, we see a distribution of number of donors available for eQTL mapping at a min-cell-threshold of 5 (also from Supplementary Table 3):

```
> summary(y$`n Donors With ≥5 Cells`)  
Min. 1st Qu. Median Mean 3rd Qu. Max.  
42.00 56.75 76.00 75.71 92.50 113.00
```

With this setting, we have at least 42 donors for all cell types, 75% of cell types have at least 56 donors, and half have at least 76 donors. Mapping this distribution of donor numbers per cell type against Fig S7 from the Cuomo et al paper we are confident that this is a sensible approach for our study to maximize detection power and control FDR (compare “power” and “empirical FDR” panels in Fig S7a with “empirical FDR” panel in Fig S7b).

The clear conclusion from applying the results from the Cuomo et al paper to this study is that we should seek to retain as many donors as possible for eQTL mapping, even if there are on average fewer cells available per donor to enable maximizing the number of donors. Overall, the results of the Cuomo et al paper show that setting a minimum cell-per-donor threshold of 5 is reasonable, and even optimal where it maximizes the number of donors included in the analysis.

A further, crucial, consideration is the coverage of cell types in a tissue for eQTL mapping. Lung is a complex tissue with many distinct cell types. We endeavor to maximize the inclusion of as many cell types, as far as is reasonable, to gain as full a picture as possible of the landscape of genetic regulation of gene expression in healthy and diseased lungs.

Setting a threshold of at least 40 donors to include a celltype in eQTL mapping and downstream analyses, we can see in the table below what a large effect the minimum cell threshold (per cell type per donor) has. With the threshold we used (≥ 5 cell) we have 38 cell types for eQTL mapping. If we set a threshold of 10 cells instead of 5, then 30 cell types are retained. This number drops starkly to 25 (≥ 20 cells), 21 (≥ 30 cells) and 16 (≥ 50 cells) as the minimum cell threshold is raised.

Number of cell types with at least n samples meeting the minimum cell number threshold for n = 30, 40, 50, 60, 70

n_cell_threshold	n30	n40	n50	n60	n70
>=5_cells	40	38	32	28	27
>=10_cells	34	30	28	22	16
>=20_cells	29	25	21	15	7
>=30_cells	25	21	16	8	6
>=50_cells	21	16	8	5	4

The plot below shows in detail how the minimum cell threshold changes the number of donors available, with one line shown per cell type.

If we insisted on a threshold of at least 50 cells per donor per cell type then we would only be able to map eQTL in 30-50% of major cell types in the lung. Using a threshold of at least 5 cells per donor allows us to map eQTL for 38 cell types, almost complete coverage of the major lung cell types.

Finally, to return to a discussion point raised in our previous response to this issue: because mashr requires a complete eQTL matrix (i.e., no missing estimates for any

eQTL for any celltype), we do run limix eQTL mapping on some genes for some cell types where there is relatively low power. The application of mashr itself, by modeling covariance and effect sharing between cell types, drastically improves eQTL detection power even in underpowered cell types. Further, after applying mashr, we apply a stricter inclusion filter where mashr-adjusted eQTL effects are only reported for genes meeting

the expression criteria for that given celltype. Thus, on balance, we think the use of mashr supports a minimum-cell-threshold of 5 cells to include as many cell types in the analysis as possible and, simultaneously, using mashr mitigates against potential issues that might otherwise arise in eQTL mapping from cell types with smaller sample sizes and therefore lower power.

In summary, we agree with the Reviewer that few single-cell eQTL studies have been done to date, which means that there is not a lot of collective experience from empirical studies to guide the choice of parameters in analysis workflows like the best choice for minimum cell threshold to set per donor per cell type. That being the case, the best information we have for making such choices comes from the Cuomo et al paper that explicitly sought to optimize single-cell eQTL mapping workflows. The results in that paper support the use of a minimum cell threshold of 5 cells, and the simulation settings used in the paper make their results directly relevant to our own study here. There is an inherent tradeoff between maximizing the number of cell types and donors for eQTL mapping and increasing the minimum number of cells per donor and cell type. On balance we think there are many good reasons to opt for a minimum cell threshold of 5 cells as this choice enables an increased inclusion of donors, and thus also cell types, which increases power for eQTL mapping and downstream mashr analysis combining eQTL results across cell types. Like the Reviewer, we expect standard approaches in the field to crystallize in the coming years. For now, we are confident that setting a minimum cell threshold of 5 is the best choice given the information we have available to us and the tradeoffs inherent in selecting the threshold.

Comment 4.

I asked the authors what strategy they used to remove Doublets. In the response letter they indicated they used Doubletfinder. However, it doesn't appear that this is described (or justified) in the methods of the main text. This is an important choice and should be described.

We apologize for the confusion around our doublet removal strategy. In our initial

response we had indicated while we have found automated double detection approaches to perform well, we have chosen a more conservative manual approach to remove doublets. We have used this approach in our previous work (<https://www.science.org/doi/full/10.1126/sciadv.aba1972>, <https://www.nature.com/articles/s41467-021-24467-0>). We have expanded upon our explanation in the methods section, as well as referenced our prior work justifying this approach. The new text now reads “We have removed doubles using a manual approach, as described previously 8,41, identifying clusters of cells that express markers from multiple lineages. Our prior work has found this method to be more conservative than automated approaches. Indeed, when applying DoubletFinder 44 to one lineage (epithelial cells), DoubletFinder recovered 8,230 doublets (3.7%), while the marker based approach identified 18,588 doublets (8.5%).”

Comment 8

The authors did not bother to assess enrichment of their eQTLs in regulatory elements at all.

We thank the Reviewer for identifying this missing analysis, and agree this is an important characterization. To this end, we have explored the genomic regions overlapping eQTL in Fig. S13 (previously S12). We have now also tested for the enrichment of eQTL among these regions by comparing the significant eQTL to a subset of non-eQTL SNPs. Non-interaction eQTL were more often found overlapping promoters (5.64-5.75% of eQTL) than int-eQTL (4.97%), the distributions of genic annotations of eQTL did not differ significantly from the null set.

We have also explored the overlap of the various classes of eQTL among all enhancers in EnhancerAtlas 2.0 (doi: 10.1093/nar/gkz980), lung tissue enhancers, and human lung epithelial cell line (Calu-3) enhancers, as well as the *cis*-regulatory elements in the Human Cell Atlas (doi: 10.1101/2023.11.13.566791). Testing for the equality of proportions overlapping enhancer annotations between eQTL and the null set, we find that multi-state sc-eQTL were more likely to be found overlapping the HCA CREs than the null set ($p=3.502E-11$). Overall, our ability to test for the enrichment of sc-eQTL is limited, as reliable enhancer annotations are not available for all relevant cell types and conditions (ILD vs. unaffected lung).

Fig. S13. Cell type-eQTL, int-eQTL, and a null set of non-eQTL SNPs annotated for genic regions. The set of non-eQTL SNPs was selected to match the total number of significant sc-eQTL and their distribution of distances to target gene transcription start sites.

Decision Letter, second revision:

Our ref: NG-A62219R1

8th Dec 2023

Dear Dr. Banovich,

Thank you for submitting your revised manuscript "Cell type-specific and disease-associated eQTL in the human lung" (NG-A62219R1). It has now been seen by the original referees and their comments are below. The reviewers find that the paper has improved in revision, and therefore we'll be happy in principle to publish it in Nature Genetics, pending minor revisions to satisfy the referees' final requests and to comply with our editorial and formatting guidelines.

We are now performing detailed checks on your paper and will send you a checklist detailing our editorial and formatting requirements soon. Please do not upload the final materials and make any

revisions until you receive this additional information from us.

Thank you again for your interest in Nature Genetics.
Please do not hesitate to contact me if you have any questions.

Best wishes,
Chiara

Chiara Anania, PhD
Associate Editor
Nature Genetics
<https://orcid.org/0000-0003-1549-4157>

Reviewer #3 (Remarks to the Author):

Generally, I am happy with the author responses to my queries. One small request I have is that the authors add some discussion on the issue of the minimum number of cells used in their pseudobulk analysis. As they articulated in their thoughtful response, they were influenced by the choice of power by including more individuals, and their experience from Cuomo et al. Clearly this is something that the field will struggle with for some time to come, and talking about the tradeoffs in the main text would be helpful for many.

Final Decision Letter:

28th Feb 2024

Dear Dr. Banovich,

I am delighted to say that your manuscript "Cell type-specific and disease-associated eQTL in the human lung" has been accepted for publication in an upcoming issue of Nature Genetics.

Due to the importance of these deadlines, we ask that you please let us know now whether you will be difficult to contact over the next month. If this is the case, we ask you provide us with the contact

information (email, phone and fax) of someone who will be able to check the proofs on your behalf, and who will be available to address any last-minute problems.

Your paper will be published online after we receive your corrections and will appear in print in the next available issue. You can find out your date of online publication by contacting the Nature Press Office (press@nature.com) after sending your e-proof corrections.

Please note that *Nature Genetics* is a Transformative Journal (TJ). Authors may publish their research with us through the traditional subscription access route or make their paper immediately open access through payment of an article-processing charge (APC). Authors will not be required to make a final decision about access to their article until it has been accepted. Find out more about Transformative Journals

Authors may need to take specific actions to achieve compliance with funder and institutional open access mandates. If your research is supported by a funder that requires immediate open access (e.g. according to Plan S principles) then you should select the gold OA route, and we will direct you to the compliant route where possible. For authors selecting the subscription publication route, the journal's standard licensing terms will need to be accepted, including <https://www.nature.com/nature-portfolio/editorial-policies/self-archiving-and-license-to-publish>. Those licensing terms will supersede any other terms that the author or any third party may assert apply to any version of the manuscript.

If you have not already done so, we invite you to upload the step-by-step protocols used in this manuscript to the Protocols Exchange, part of our on-line web resource, natureprotocols.com. If you complete the upload by the time you receive your manuscript proofs, we can insert links in your article that lead directly to the protocol details. Your protocol will be made freely available upon publication of your paper. By participating in natureprotocols.com, you are enabling researchers to more readily reproduce or adapt the methodology you use. [Natureprotocols.com](http://natureprotocols.com) is fully searchable, providing your protocols and paper with increased utility and visibility. Please submit your protocol to <https://protocolexchange.researchsquare.com/>. After entering your nature.com username and password you will need to enter your manuscript number (NG-A62219R2). Further information can be found at <https://www.nature.com/nature-portfolio/editorial-policies/reporting-standards#protocols>

Sincerely,
Chiara

Chiara Anania, PhD
Associate Editor
Nature Genetics
<https://orcid.org/0000-0003-1549-4157>